# Rényi entropies for one-dimensional quantum systems with mixed boundary conditions

Benoit Estienne, Yacine Ikhlef, Andrei Rotaru

Sorbonne Université, CNRS, Laboratoire de Physique Théorique et Hautes Énergies, LPTHE, F-75005 Paris, France

January 6, 2023

## Abstract

We present a general method for calculating Rényi entropies in the ground state of a one-dimensional critical system with mixed open boundaries, for an interval starting at one of its ends. In the conformal field theory framework, this computation boils down to the evaluation of the correlation function of one twist field and two boundary condition changing operators in the cyclic orbifold. Exploiting null-vectors of the cyclic orbifold, we derive ordinary differential equations satisfied by these correlation functions. In particular we obtain an explicit expression for the second Rényi entropy valid for any diagonal minimal model, but with a particular set of mixed boundary conditions. In order to compare our results with numerical data for the Ising and three-state Potts critical chains, we also identify and compute the leading finite size corrections.

# 1   Introduction

The understanding of quantum entanglement has proved to be a research topic of continued and central interest for physicists working in domains as diverse as high energy physics, condensed matter theory and quantum information. Entanglement measures have turned out to be useful diagnostic tools for tensor network algorithms, quantities of interest for the AdS/CFT correspondence, and, most relevantly for the present work, a powerful tool for probing the physics of quantum many-body systems. With respect to the latter, the study of entanglement has proved crucial to the study of phase transitions in one dimensional quantum systems, by allowing their detection and the characterization of their critical exponents and corresponding central charge [1–4]. Important applications of entanglement are found in higher dimensions too. We mention, for two-dimensional systems, the establishment of intrinsic topological order and various anyonic quantum dimensions [5, 6] and the detection and counting of critical Dirac fermions [7–10]. Finally, entanglement can also be used, in two [11–15] or higher dimensions [16, 17] to reveal gapless interface modes.

The basic setup is as follows: we consider a quantum system in a pure state $|\Psi\rangle$, and a spatial bipartition of said system into two complementary subregions $A$ and $B$. The entanglement between them is then encoded in the reduced density matrix $\rho_A = \mathrm{Tr}_B |\Psi\rangle\langle\Psi|$ and it can be quantified through entanglement measures, such as the *Rényi entanglement entropies* [18–22]

$$S_n(A) = \frac{1}{1-n} \log \mathrm{Tr}_A\left(\rho_A^n\right) , \tag{1.1}$$

and in particular the $n \to 1$ case corresponding to the well-known *von Neumann entropy*:

$$S(A) = -\mathrm{Tr}_A\left(\rho_A \log \rho_A\right) . \tag{1.2}$$

While the focus on entanglement entropies has been mostly theoretical, in recent years experimental proposals as well as actual experiments have been designed to measure them [23–28].

For strongly correlated quantum systems, the theoretical computation of entanglement entropies is a technically challenging endeavour. However, if these systems are one-dimensional and critical, the formidable toolbox of two-dimensional Conformal Field Theory (CFT) is available to tackle such computations. The calculations of entanglement entropies through such methods rests on two crucial insights. The first insight is that, for integer values of $n$, and a subsystem $A = \cup_i[u_i, v_i]$ built as the union of some disjoint intervals, the moments of the reduced density matrix $\mathrm{Tr}_A\left(\rho_A^n\right)$ can be expressed as the partition function of an $n$-sheeted Riemann surface with conical singularities corresponding to the endpoints of the intervals $[u_i, v_i]$ [2, 29]. Such partition functions have been evaluated, with significant toil, for free theories and some special cases of interacting models [4, 30–38]. In general, however, a second insight is needed to make progress: the replication of the *spacetime* of the theory can be "exchanged" for the replication of the *target space* of the CFT [39–41]. Such a construction, known in the literature as the *cyclic orbifold CFT* [40], is built from the permutation symmetric product of $n$ copies of the original CFT (referred to as *the mother CFT*), by modding out the discrete subgroup $\mathbb{Z}_n$ of cyclic permutations. In this framework, the conical singularities of the mother CFT defined on the replicated surface are accounted for by insertions of *twist fields* [39] in cyclic orbifold correlators. Thus, by computing correlators of twist operators, one can evaluate $\mathrm{Tr}_A\left(\rho_A^n\right)$ for a variety of setups. To give a few examples, one can easily adapt the twist field formalism to encode modified initial conditions around the branch points [42], which is fitting for computations of more refined entanglement measures such as the symmetry-resolved entanglement entropy [43–48] or for explorations of entanglement in non-unitary systems [42, 49]. Arguably

the most renowned result obtained in this framework is [1, 2, 29, 50–52]

$$S_n(\ell) \underset{\ell \to \infty}{\sim} \frac{c}{6} \frac{n+1}{n} \log \ell \,, \tag{1.3}$$

which gives the *universal* asymptotic behaviour for the ground state entanglement entropy of an interval of length $\ell$ in an infinite system (with $c$ the central charge of the critical system).

In this article, we consider the Rényi entanglement entropy in an open system with *mixed boundary conditions*, when the subregion $A$ is a single interval *touching the boundary* – we take the boundary condition (BC) at one end of the chain to be different from the BC at the other end (see Figure 1). In the scaling limit, such an open critical system is described by a Boundary Conformal Field Theory (BCFT), with a well understood [53–56] correspondence between the chiral Virasoro representations and the *conformal boundary conditions* allowed by the theory, and an algebra of boundary operators that interpolate between them.

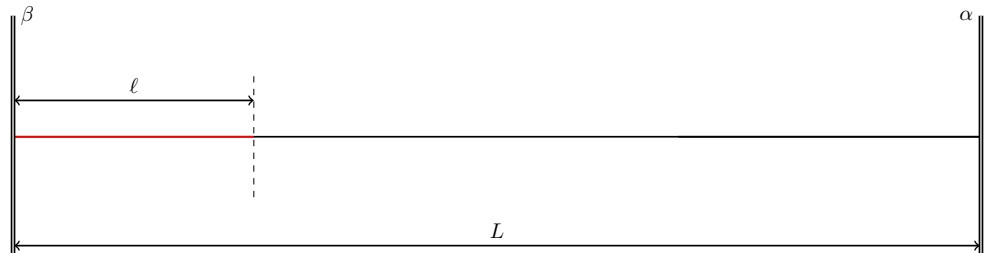

Figure 1: An interval of length $\ell$ in a 1d critical chain with mixed BC ($\alpha\beta$) and length $L$.

The more accessible setup of an interval touching one of two *identical* boundaries has been thoroughly analysed using either conformal field theory methods [2, 52, 57–60] or exact free fermion techniques [61–63]. Such configurations are also well-handled numerically, through density-matrix renormalization group (DMRG) techniques [64–67]. In that setup, the subsystem $A$ is at the end of a finite system with the same boundary condition $\alpha$ on both sides. The computation of the Rényi entanglement entropies rests on the evaluation of a twist one-point function on the upper half-plane. Such a correlation function is straightforwardly fixed by conformal invariance, and as a consequence the entanglement entropy exhibits a simple dependence on the interval and system sizes. Explicitly, in the case of an interval of length $\ell$ at the end of a system of size $L$, one finds the leading universal behaviour [2]:

$$S_n(\ell) \sim \frac{c}{12} \frac{n+1}{n} \log \left[ \frac{2L}{\pi a} \sin \left( \frac{\pi \ell}{L} \right) \right] + \log g_\alpha \,, \tag{1.4}$$

where $a$ is the lattice spacing and $g_\alpha$ is the *universal boundary entropy* [68] associated to the boundary condition $\alpha$.

When one studies systems with mixed BC, at the level of the BCFT one has to introduce *boundary condition changing operators* (BCCOs), and thus the corresponding correlators are more complicated. The core idea of this framework is that the singular behaviour associated to the change in boundary conditions, can be encoded in the form of operators placed on the boundary, that interpolate between regions of different BC $\alpha \neq \beta$. Thus, to compute the Rényi entropy $S_n$ in this setup, we will evaluate *three-point functions* with one twist operator and two BCCO insertions. Such setups have already been studied for the Ising and XX chains in [57], at the level of the CFT on the replicated surface, and rely on the knowledge of relatively simple closed form expressions for the $2n$-point correlator of BCCOs on the unit disk for their calculations. However, such knowledge is the exception, rather than the norm, for generic BCFTs.

In this work, we present a general method to compute such twist correlators functions with mixed BCs. The most technically demanding part of this framework is finding Ordinary Differential Equations (ODEs) that the correlators satisfy. According to Cardy's doubling trick [53], in the half-plane geometry, the three-point functions of interest obey the same Ward identities as a four-point conformal block with the corresponding operators, where the bulk twist operator $\sigma(z, \bar{z})$ is replaced by the insertion of $\sigma(z)\sigma^\dagger(\bar{z})$. Thus, in an adaptation of the method of [42], we can derive a differential equation by combining knowledge of the null-vector conditions obeyed by the twisted and untwisted fields under the symmetry algebra of the cyclic orbifold [40] with the derivation of well-chosen Ward identities obtained from current insertions in the correlators of interest. The final ingredient is the determination of a subset of the (bulk and boundary) structure constants of the cyclic orbifold BCFT, which fix the specific linear combination of solutions of the differential equation that gives the sought correlator.

We have illustrated this approach with a variety of BCFT setups, that share a common assumption: in the mother CFT, the mixed boundary conditions $(\beta\alpha)$ are implemented by a BCCO which is degenerate at level two under the Virasoro algebra.

With this restriction, in the $\mathbb{Z}_2$ orbifold of a generic BCFT, we have derived a second-order and a fourth order ODE, respectively for the *bare* and *composite*[1] twist correlator. Under the same restrictions, in the $\mathbb{Z}_3$ orbifold of a generic BCFT, we have determined a third-order ODE for the bare twist correlator. We have also worked out, for the case of the $\mathbb{Z}_2$ and $\mathbb{Z}_3$ cyclic orbifolds of the Ising BCFT, a variety of lower-order ODEs. The latter calculations were found compatible with the results of [57], and have been tested against numerical results for the critical Ising chain, for all possible combinations of BCs, to excellent agreement. Finally, we have also considered the $\mathbb{Z}_2$ orbifold of the three-state Potts model, and compared it against lattice data for the critical three-state Potts model with states $\{R, G, B\}$, with less accurate, but consistent results. We quote here the leading behaviour of the second Rényi entropy of the critical three-state Potts chain of size $L$ for mixed fixed $R$ and *restricted GB* boundary conditions:

$$S_2^{(R,GB)}(\ell) \sim \frac{c_{Potts}}{8} \log \frac{2L}{\pi a} \sin\left(\frac{\pi\ell}{L}\right) + \log g_R - \log\left[\eta^{-2h_{12}} {}_2F_1\left(-8/5, -9/10; -9/5 \mid 1 - \eta\right)\right]$$
(1.5)

with $\eta = e^{2\pi i\ell/L}$, $c_{Potts} = 4/5$ the central charge, $h_{12} = 2/5$ the scaling dimension of the BCCO, and $g_R = [(5 - \sqrt{5})/30]^{1/4}$ the ground state degeneracy associated to the fixed $R$ BC [69].

The expression (1.5) is, in fact, only a particular case of a more general result obtained in this paper, which applies to any critical system described by a BCFT based on a minimal model $\mathcal{M}(p, p')$ with mixed conformal BC $(\alpha, \beta)$ chosen such that the most relevant BCCO interpolating between them is $\psi_{1,2}^{(\alpha\beta)}$ and there is no boundary operator $\psi_{1,3}^{(\beta\beta)}$ allowed in the theory. Under these conditions, the second Rényi entropy of an interval $A = [0, \ell]$, in a finite system of size $L$ touching the boundary $\beta$ is:

$$S_2^{(\alpha,\beta)}(\ell) \sim \frac{c}{8} \log \frac{2L}{\pi a} \sin\left(\frac{\pi\ell}{L}\right) + \log g_\beta - \log\left[\eta^{-2h_{12}} {}_2F_1\left(2 - 3\frac{p}{p'}, 3/2 - \frac{p}{p'}; 3 - 4\frac{p}{p'} \mid 1 - \eta\right)\right]$$
(1.6)

where $c$, $g_\beta$ and $h_{12}$ generalize the notation in (1.5).

In both (1.5) and (1.6), one should keep in mind, especially for the purpose of numerical studies, that the hypergeometric function in the third term of these equations converges inside the unit circle centred at $\eta = 1$, which only overlaps with the subinterval $Arg(\eta) \in (0, \pi/3) \cup$

---

[1] obtained by fusing the bare twist operator with an untwisted operator $\phi$.

$(5\pi/3, 2\pi)$ of the parameter space of interest $Arg(\eta) \in [0, 2\pi]$. Thus, to evaluate the expressions for $Arg(\eta) \in (\pi/3, 5\pi/3)$, it is necessary to analytically continue the third term to this range.

We give here the outline of the article. In Section 2, we give a review of the cyclic orbifold construction, with a focus on its implementation on the upper-half plane. We discuss in this section the bulk and boundary operator algebra, and show how some orbifold bulk and boundary structure constants can be expressed in terms of mother BCFT quantities by unfolding and factorizing arguments. We dedicate Section 3 to the derivation of ODEs for the different setups described above. On top of the announced derivations involving orbifold Ward identities, we also use the results on the fusion rules of the $\mathbb{Z}_N$ cyclic orbifold of [70] and some mathematical facts about the hypergeometric differential equation, to derive low-order differential equations for the Ising case. Section 4 contains a comparison of our analytical results with lattice data, for both the Ising and three-state Potts critical chains. Finally, we have relegated the more technical derivations to the Appendix, to avoid congesting the logical flow of the paper.

# 2 The cyclic orbifold

In this section, we will present the construction of the cyclic orbifold BCFT on the upper half-plane $\mathbb{H}$. After reviewing a few essential features of the $\mathbb{Z}_N$ orbifold on the Riemann sphere, we will discuss conformal boundary conditions, boundary operators as well as bulk-boundary and boundary-boundary operator algebras.

## 2.1 The cyclic orbifold on the Riemann sphere

To build a cyclic orbifold CFT, one starts from any mother CFT $\mathcal{M}$ and constructs the tensor product theory $\mathcal{M}^{\otimes N}$. Then one considers all the $\mathbb{Z}_N$ equivalent ways of connecting the copies of the product theory, which creates $N$ different sectors, each with its corresponding operator families and labelled by a $\mathbb{Z}_N$ *twist charge* $[k]$. The spectrum of the cyclic orbifold $\mathcal{M}_N$ is then built as a reunion of the operator families from all the sectors $[k]$.

### 2.1.1 Symmetry algebra and operator content

In $\mathcal{M}_N$, each copy $a$ of the mother CFT carries the components of the stress-energy tensor $T_a(z), \bar{T}_a(\bar{z})$. We define the discrete Fourier modes of these currents as

$$T^{(r)}(z) = \sum_{a=0}^{N-1} \omega^{ar} \, T_a(z) \,, \qquad \bar{T}^{(r)}(\bar{z}) = \sum_{a=0}^{N-1} \omega^{ar} \, \bar{T}_a(\bar{z}) \,, \qquad (2.1)$$

where $r$ is considered modulo $N$, and we have used the notation $\omega = \exp(2i\pi/N)$. They satisfy the OPEs

$$T^{(r)}(z)T^{(s)}(w) = \frac{\delta_{r+s,0} \, Nc/2}{(z-w)^4} + \frac{2T^{(r+s)}(w)}{(z-w)^2} + \frac{\partial T^{(r+s)}(w)}{z-w} + \mathrm{reg}_{z\to w} \,,$$

$$\bar{T}^{(r)}(\bar{z})\bar{T}^{(s)}(\bar{w}) = \frac{\delta_{r+s,0} \, Nc/2}{(\bar{z}-\bar{w})^4} + \frac{2\bar{T}^{(r+s)}(\bar{w})}{(\bar{z}-\bar{w})^2} + \frac{\partial \bar{T}^{(r+s)}(\bar{w})}{\bar{z}-\bar{w}} + \mathrm{reg}_{\bar{z}\to \bar{w}} \,, \qquad (2.2)$$

where the Kronecker symbols $\delta_{r+s,0}$ are understood modulo $N$. The symmetric modes $T^{(0)}(z)$ and $\bar{T}^{(0)}(\bar{z})$ are the components of the stress-energy tensor of $\mathcal{M}_N$ with central charge $Nc$,

whereas the other Fourier modes $T^{(r)}(z), \bar{T}^{(r)}(\bar{z})$ with $r \neq 0$ should be regarded as additional conserved currents. Altogether, these Fourier modes encode an extended conformal symmetry. The modes associated to these currents are defined in the usual way through:

$$L_m^{(r)} = \frac{1}{2i\pi} \oint dz\, z^{m+1}\, T^{(r)}(z)\,,$$
$$\bar{L}_m^{(r)} = \frac{1}{2i\pi} \oint d\bar{z}\, \bar{z}^{m+1}\, \bar{T}^{(r)}(\bar{z})\,. \tag{2.3}$$

In the sector of twist charge $[k]$ one has the following mode decompositions

$$T^{(r)}(z) = \sum_{m \in -kr/N + \mathbb{Z}} z^{-m-2} L_m^{(r)}$$
$$\bar{T}^{(r)}(\bar{z}) = \sum_{m \in +kr/N + \mathbb{Z}} \bar{z}^{-m-2} \bar{L}_m^{(r)} \tag{2.4}$$

and the commutation relations

$$\left[ L_m^{(r)}, L_n^{(s)} \right] = (m-n) L_{m+n}^{(r+s)} + \frac{Nc}{12} m(m^2-1)\, \delta_{m+n,0}\, \delta_{r+s,0}\,,$$
$$\left[ \bar{L}_m^{(r)}, \bar{L}_n^{(s)} \right] = (m-n) \bar{L}_{m+n}^{(r+s)} + \frac{Nc}{12} m(m^2-1)\, \delta_{m+n,0}\, \delta_{r+s,0}\,. \tag{2.5}$$

Hermitian conjugation of the modes acts as:

$$\left( L_n^{(r)} \right)^\dagger = L_{-n}^{(-r)}\,, \qquad \left( \bar{L}_n^{(r)} \right)^\dagger = \bar{L}_{-n}^{(-r)}\,. \tag{2.6}$$

Orbifold *primary operators* are, by definition, annihilated by the action of all the positive modes of $\mathrm{OVir} \otimes \overline{\mathrm{OVir}}$. Descendant operators with respect to this algebra are constructed by the action of the negative $m$ modes. We establish the notation for descendants of a scaling (primary or not) operator $\mathcal{O}$:

$$\left( L_m^{(r)} \cdot \mathcal{O} \right)(z, \bar{z}) := \frac{1}{2i\pi} \oint_{C_z} dw\, (w-z)^{m+1}\, T^{(r)}(w) \mathcal{O}(z, \bar{z})\,,$$
$$\left( \bar{L}_m^{(r)} \cdot \mathcal{O} \right)(z, \bar{z}) := \frac{1}{2i\pi} \oint_{C_z} d\bar{w}\, (\bar{w}-\bar{z})^{m+1}\, \bar{T}^{(r)}(\bar{w}) \mathcal{O}(z, \bar{z})\,, \tag{2.7}$$

where the contour $C_z$ encloses the point $z$.

It will be useful to work with the primary operator spectrum with respect to the *neutral subalgebra* $A \otimes \bar{A}$ generated by the algebra elements

$$L_{m_1}^{(r_1)} \ldots L_{m_p}^{(r_p)} \quad \text{and} \quad \bar{L}_{m_1}^{(r_1)} \ldots \bar{L}_{m_p}^{(r_p)}\,, \qquad \text{with } r_1 + \cdots + r_p = 0 \mod N\,. \tag{2.8}$$

One can classify all $\mathbb{Z}_N$-symmetric operators of $\mathcal{M}_N$ into representations of $A \otimes \bar{A}$. This organization, described in detail by the authors of the present work in [70], distinguishes between three types of operators. First, we have identified the *untwisted non-diagonal operators* $\Phi_{[j_1 \ldots j_N]}$. These operators are built from $\mathbb{Z}_N$-symmetrized combinations of products of mother CFT primary operators $\phi_j$ (with $j = 1$ referring to the identity operator $\mathbf{1}$):

$$\Phi_{[j_1 \ldots j_N]} := \frac{1}{\sqrt{N}} \sum_{a=0}^{N-1} (\phi_{j_{1+a}} \otimes \cdots \otimes \phi_{j_{N+a}})\,, \tag{2.9}$$

in which at least one pair satisfies $j_i \neq j_k$. Its conformal dimension is given by $h_{[j_1 \ldots j_N]} = \sum_s h_{j_s}$.

The second type of primary operators under the neutral algebra are the *untwisted diagonal fields* $\Phi_j^{(r)}$, where the Fourier replica index $r$ takes values in $\mathbb{Z}_N$ . The $r = 0$ diagonal fields are defined to be:

$$\Phi_j^{(0)} = \Phi_j := \phi_j \otimes \cdots \otimes \phi_j \,, \tag{2.10}$$

while for $r \neq 0$, they are constructed as:

$$\Phi_j^{(r)} := \frac{1}{2Nh_j} L_{-1}^{(r)} \bar{L}_{-1}^{(-r)} \cdot \Phi_j \,, \qquad \mathbf{1}^{(r)} := \frac{2}{Nc} L_{-2}^{(r)} \bar{L}_{-2}^{(-r)} \cdot \Phi_{\mathbf{1}} \,, \tag{2.11}$$

The conformal dimension of a diagonal operator $\Phi_j^{(r)}$ is then generically given by

$$h_j^{(r)} = Nh_j + \delta_{r,0} \left(1 + \delta_{j,1}\right) \tag{2.12}$$

We should note that the diagonal operators with $r = 0$ and the non-diagonal operators are also primary under $\mathrm{OVir} \otimes \overline{\mathrm{OVir}}$.

Finally, we have to consider twist operators, which come in distinct flavours. For the purposes of this paper, we will mostly work with twist operators with Fourier replica index $r = 0$. Thusly, just as for the diagonal fields, we will drop this specification when the context heavily implies it, to decongest the notation.

We first consider the ubiquitous bare twist operators [2, 30, 71, 39] which are denoted in our conventions $\sigma^{[k]} = \sigma_{\mathbf{1}}^{[k]}$, or, in light notation, $\sigma = \sigma^{[1]}$ and $\sigma^\dagger = \sigma^{[-1]}$ . We have also the composite twist fields $\sigma_j^{[k]}$, which can be defined through point-splitting as in [49]:

$$\sigma_j^{[k]}(z, \bar{z}) := \mathcal{A}_j \lim_{\epsilon \to 0} \left[ \epsilon^{2(1-N^{-1})h_j} \Phi_{[j,\mathbf{1},\ldots,\mathbf{1}]}(z + \epsilon, \bar{z} + \bar{\epsilon}) \cdot \sigma^{[k]}(z, \bar{z}) \right] \,, \tag{2.13}$$

where the constant $\mathcal{A}_j = N^{-2(1-N^{-1})h_j - 1/2}$ ensures that non-vanishing two-point functions of twist operators are normalized to one. If $N$ and $k$ are coprime, the conformal dimension of the bare twist operator is

$$h_\sigma = \frac{c}{24} \left( N - \frac{1}{N} \right) \,, \tag{2.14}$$

while for composite twist operators one has:

$$h_{\sigma_j} = h_\sigma + \frac{h_j}{N} \,. \tag{2.15}$$

Having established the primary operator spectrum of the orbifold, we will now review how the null-vectors of the diagonal and twisted fields in $\mathcal{M}_N$ are inferred from the ones of the mother theory $\mathcal{M}$.

### 2.1.2 Null vectors for untwisted operators

Let us consider a generic mother CFT $\mathcal{M}$, with central charge

$$c = 1 - \frac{6(1-g)^2}{g} \,, \qquad 0 < g \leq 1 \,. \tag{2.16}$$

The conformal dimensions of degenerate primary operators are given by the Kac formula

$$h_{rs} = \frac{(r - sg)^2 - (1 - g)^2}{4g} \,, \tag{2.17}$$

where $r, s$ are positive integers. The corresponding operator $\phi_{rs}$ is degenerate at level $rs$. If the parameter $g$ is rational, i.e. $g = p/p'$ with coprime $p$ and $p'$, then the set of operators $\phi_{rs}$ with $1 \leq r \leq p - 1$ and $1 \leq s \leq p' - 1$ generates a closed operator algebra, and the related CFT is the minimal model $\mathcal{M}_{p,p'}$. While we do employ this parametrization extensively, in the present work we will consider a more generic mother CFT, and we *do not assume* that it is a minimal model—unless explicitly indicated.

Consider the situation when the mother CFT includes the degenerate operator $\phi_{12}$, with null-vector condition

$$\left( L_{-2} - \frac{1}{g} L_{-1}^2 \right) \phi_{12} = 0. \tag{2.18}$$

In the untwisted sector of the orbifold CFT, we have

$$L_n^{(r)} = \sum_{a=1}^{N} e^{2i\pi r a/N} \left( 1 \otimes \ldots 1 \otimes \underset{(a-\text{th})}{L_n} \otimes 1 \otimes \ldots 1 \right), \qquad n \in \mathbb{Z}, \tag{2.19}$$

and the diagonal untwisted operator associated to $\phi_{12}$ is

$$\Phi_{12} = \phi_{12} \otimes \cdots \otimes \phi_{12}. \tag{2.20}$$

Using an inverse discrete Fourier transform, one easily finds, for any $r \in \mathbb{Z}_N$,

$$\left[ L_{-2}^{(r)} - \frac{1}{Ng} \sum_{s=0}^{N-1} L_{-1}^{(s)} L_{-1}^{(r-s)} \right] \cdot \Phi_{12} = 0. \tag{2.21}$$

When inserted into a correlation function, the modes $L_m^{(0)}$ act as linear differential operators. The treatment of the modes $L_m^{(r)}$ with $r \neq 0$ introduces an additional difficulty, that we will address case by case, with the help of orbifold Ward identities.

### 2.1.3 The induction procedure

The null-vectors of the mother CFT also determine the null vector conditions on twist operators in $\mathcal{M}_N$, through the *induction procedure* [40].

In the present work, we shall only be concerned with the twist sectors with charges $[\pm 1]$. In the notations of [70], induction can be expressed in terms of a norm-preserving, invertible linear map $\Theta$ from the Hilbert space of the mother CFT to that of the twist sector [1], defined by

$$\Theta |\phi\rangle = |\sigma_\phi\rangle, \qquad \Theta L_m \Theta^{-1} = N \left( L_{m/N}^{(-m)} - h_\sigma \, \delta_{m0} \right), \tag{2.22}$$

where $\phi$ is any primary operator in the mother CFT, and $\sigma_\phi$ is the associated composite twist operator in the orbifold CFT.

The simplest application to null-vectors is the case of the identity:

$$L_{-1} \cdot \mathbf{1} = 0 \qquad \Rightarrow \qquad L_{-1/N}^{(1)} \cdot \sigma = 0. \tag{2.23}$$

For a degenerate operator at level two, applying the induction map on (2.18) yields

$$\left[ L_{-2/N}^{(2)} - \frac{N}{g} (L_{-1/N}^{(1)})^2 \right] \cdot \sigma_{12} = 0. \tag{2.24}$$

The corresponding null-vector conditions for the operators $\sigma^\dagger$ and $\sigma_{12}^\dagger$ are easily obtained by conjugation.

## 2.2 The cyclic orbifold on the upper half plane

To construct the cyclic orbifold BCFT, we will work on the upper half-plane $\mathbb{H}$, with the boundary along the real axis. We parametrize $\mathbb{H}$ by $z = x + iy$ with $x \in \mathbb{R}$ and $y > 0$, and we impose the gluing condition on the boundary for the stress-energy tensor components:

$$T^{(0)}(x) = \bar{T}^{(0)}(x) \quad \text{for} \quad x \in \mathbb{R}, \tag{2.25}$$

which ensures that the boundary is conformal i.e., preserves a copy of the Virasoro algebra [72]. The $\mathbb{Z}_N$ orbifold, however, has an extended symmetry, and we must choose if and how the components of the additional currents $T^{(r \neq 0)}$ are glued at the boundary. Our usage of the replica trick provides a clear indication for these choices: since we are considering $N$ copies of the *same* mother BCFT, we must impose the gluing condition $T_a(x) = \bar{T}_a(x)$ on each of them. By taking the Fourier transform of this relation, we find that in the orbifold CFT we are effectively imposing:

$$T^{(r)}(x) = \bar{T}^{(r)}(x) \quad \text{for} \quad x \in \mathbb{R}, \tag{2.26}$$

for all the discrete Fourier modes of the stress-energy tensor components defined in (2.1). This implies that the boundary preserves a full copy of the OVir algebra.

By the same reasoning on CFT replicas, the orbifold boundary states we are interested in correspond to having the same conformal BC on the $N$ copies of the mother CFT. They are simply given by $|\alpha\rangle^{\otimes N}$ and $|\beta\rangle^{\otimes N}$.

On the upper half-plane, we will set the conformal BC $\alpha$ on the positive real axis $x > 0$ and the conformal BC $\beta$ on $x < 0$. To implement such mixed conformal BC in a BCFT, we will have to work with the formalism of *boundary condition changing operators* [53]. These operators, restricted to live on the boundary, are placed at the points of suture of regions of different BC. The full operator algebra of a BCFT is then formed by considering the OPEs between both BCCOs and bulk operators, as detailed in Appendix A. For a given pair of conformal BCs $(\alpha, \beta)$, there can be several primary BCCOs implementing the change $\alpha \to \beta$: we denote such an operator $\psi_h^{(\alpha\beta)}$, where $h$ specifies its conformal dimension. The most relevant BCCO implementing $\alpha \to \beta$ is simply referred to as $\psi^{(\alpha\beta)}$.

In the $\mathbb{Z}_N$ orbifold CFT, we will be concerned with the calculation of correlators with insertions of *diagonal BCCOs*, namely :

$$\Psi_h^{(\alpha\beta)} = \underbrace{\psi_h^{(\alpha\beta)} \otimes \cdots \otimes \psi_h^{(\alpha\beta)}}_{N \text{ times}}. \tag{2.27}$$

Then, orbifold correlators with mixed BC are obtained by inserting the most relevant diagonal BCCO:

$$\langle \mathcal{O}_1(z_1, \bar{z}_1) \ldots \mathcal{O}_n(z_n, \bar{z}_n) \rangle_{\mathbb{H}}^{\alpha\beta} = \langle \Psi^{(\alpha\beta)}(\infty) \, \mathcal{O}_1(z_1, \bar{z}_1) \ldots \mathcal{O}_n(z_n, \bar{z}_n) \Psi^{(\beta\alpha)}(0) \rangle_{\mathbb{H}}. \tag{2.28}$$

By Cardy's doubling trick [72], [54], such $(n+2)$-point correlators satisfy the same Ward identities as any of the $(2n+2)$-point conformal blocks on the Riemann sphere $\mathbb{C}$ with external operators

$$\Phi(\infty), \mathcal{O}_1(z_1), \overline{\mathcal{O}}_1(\bar{z}_1), \ldots, \mathcal{O}_n(z_n), \overline{\mathcal{O}}_n(\bar{z}_n), \Phi(0), \tag{2.29}$$

where $\overline{\mathcal{O}}_i(\bar{z})$ is the antiholomorphic counterpart of $\mathcal{O}_i(z)$, and $\Phi(z)$ is the holomorphic part of the diagonal primary operator defined in (2.10), with the conformal dimension of $\Psi^{(\alpha\beta)}$. In more precise terms, $\overline{\mathcal{O}}_i$ is the operator conjugate to $\mathcal{O}_i$ with respect to the symmetry algebra

preserved by the boundary [73]. For $\mathbb{Z}_N$ twist operators, conjugation acts as $\bar{\sigma}_i = \sigma_i^\dagger$ [39], so that the one-twist function

$$\langle \sigma_i(z, \bar{z}) \rangle_{\mathbb{H}}^{(\alpha\beta)} = \langle \Psi^{(\alpha\beta)}(\infty) \sigma_i(z, \bar{z}) \Psi^{(\alpha\beta)}(0) \rangle_{\mathbb{H}} \tag{2.30}$$

satisfies the same Ward identities as the functions $\bar{z}^{-2h_{\sigma_i}} \times \mathcal{F}_k(z/\bar{z})$, where $\mathcal{F}_k$ is the rescaled conformal block:

$$\mathcal{F}_k(\eta) = \begin{array}{c} \Phi(\infty) \quad\quad \Phi(0) \\ \diagdown\quad\Phi_k\quad\diagup \\ \diagup\quad\quad\diagdown \\ \sigma_i(1) \quad\quad \sigma_i^\dagger(\eta) \end{array} = \langle \Phi | \sigma_i(1) \mathcal{P}_k \sigma_i^\dagger(\eta) | \Phi \rangle \,, \tag{2.31}$$

$\mathcal{P}_k$ is the projector onto the $(A \otimes \bar{A})$-module of $\Phi_k$, and $\{\Phi_k\}$ is the set of allowed intermediary untwisted (diagonal or not) primary operators. In the following, it will also be necessary to consider the conformal blocks in the channel $\eta \to 1$, namely

$$\widetilde{\mathcal{F}}_\ell(\eta) = \begin{array}{c} \sigma_i^\dagger(\eta) \quad\quad \Phi(0) \\ \diagdown\quad\Phi_\ell\quad\diagup \\ \diagup\quad\quad\diagdown \\ \sigma_i(1) \quad\quad \Phi(\infty) \end{array} = \langle \Phi | \Phi(1) \mathcal{P}_\ell \sigma_i^\dagger(1 - \eta) | \sigma_i \rangle \,. \tag{2.32}$$

Using the Ward identities implied by the OVir algebra (discussed in detail in Section 3) for these functions, together with the null-vectors of the previous section, will allow us to extract an ODE that the functions (2.31–2.32) satisfy.

To understand the structure of the conformal blocks, we also need to define *non-diagonal* BCCOs, paralleling (2.9):

$$\Psi^{(\alpha\beta)}_{[j_1 \dots j_N]} := \frac{1}{\sqrt{N}} \sum_{\ell=0}^{N-1} (\psi^{(\alpha\beta)}_{j_{1+\ell}} \otimes \cdots \otimes \psi^{(\alpha\beta)}_{j_{N+\ell}}) \,, \tag{2.33}$$

where the $1/\sqrt{N}$ factor ensures that the non-vanishing two-point functions of these BCCOs are normalized to one. These operators appear in the OPE of the diagonal boundary fields $\Psi^{(\alpha\beta)}_j$, and thus their conformal dimension determines the leading singular behaviour of the conformal blocks. Naturally, we should now discuss the operator algebra of the $\mathbb{Z}_N$ orbifold BCFT.

## 2.3 Operator algebra of the cyclic orbifold BCFT

In the orbifold BCFT, the operator algebra consists of OPEs of three types. First, there is the operator subalgebra of bulk operators, inherited from the $\mathbb{Z}_N$ orbifold CFT on $\mathbb{C}$. We shall not directly use the structure constants of this subalgebra in this work, but they have been discussed in [40, 42, 71, 74]. The second type of OPE we need to consider in the orbifold BCFT, is the bulk-boundary OPE which encapsulates the singular behaviour of a bulk field as it approaches a conformal boundary. In our calculations, we will only need to work with the OPEs of primary twist operators $\sigma_i(z, \bar{z})$ as they are sent towards a conformal boundary $\alpha$:

$$\sigma_i(x, y) \underset{y \to 0}{\sim} \sum_j \mathcal{A}^{(\alpha)}_{\sigma_i, \Psi_j} (2y)^{h_j - 2h_{\sigma_i}} \Psi^{(\alpha\alpha)}_j(x) \tag{2.34}$$

where the sum runs over all the boundary operators $\Psi^{(\alpha\alpha)}_j$ and their descendants under OVir, and we have denoted the *bulk-boundary structure constants* by $\mathcal{A}^{(\alpha)}_{\sigma_i, \Psi_j}$. Finally, we need to

consider the OPEs of orbifold boundary operators. For generic diagonal BCCOs, this takes the form

$$\Psi_{i_1}^{(\alpha\beta)}(x_1)\Psi_{i_2}^{(\beta\gamma)}(x_2) \underset{x_1 \to x_2}{\sim} \sum_j \mathcal{B}_{\Psi_{i_1},\Psi_{i_2}}^{(\beta\gamma)\Psi_j}(x_1-x_2)^{-h_{i_1}-h_{i_2}+h_j}\Psi_j^{(\alpha\gamma)}(x_2) \tag{2.35}$$

with the index $j$ running over all the orbifold BCCOs interpolating between the conformal boundary conditions $\alpha$ and $\gamma$. We have denoted the *boundary-boundary structure constants* by $\mathcal{B}_{\Psi_{i_1},\Psi_{i_2}}^{(\alpha\beta\gamma)\Psi_j}$. To calculate the structure constants of the OPEs that are relevant for the present work, we will need to use factorization and unfolding arguments for the correlator that determine them, along the lines of [70], [75] and [71].

### 2.3.1 Calculation of boundary-boundary structure constants

Let us consider the calculation of boundary-boundary structure constants of the type $\mathcal{B}_{\Psi_*,\Psi_j}^{(\beta\beta\alpha)\Psi_k}$, where $\Psi_*$ denotes a generic untwisted orbifold primary BCCO. We can express this as a three-point function on the upper half-plane $\mathbb{H}$:

$$\mathcal{B}_{\Psi_*,\Psi_j}^{(\beta\beta\alpha)\Psi_k} = \langle \Psi_k^{(\alpha\beta)}(\infty)\Psi_j^{(\beta\beta)}(1)\Psi_*^{(\beta\alpha)}(0)\rangle_{\mathbb{H}}. \tag{2.36}$$

Since there are no twist insertions in the above correlator, it just factorizes into a linear combination of products of mother BCFT three-point functions. Let us first consider the case of a diagonal BCCO, with $\Psi_*^{(\beta\alpha)} = \Psi_i^{(\beta\alpha)}$. Then, the orbifold correlator factorizes into mother CFT three-point functions as:

$$\mathcal{B}_{\Psi_i,\Psi_j}^{(\beta\beta\alpha)\Psi_k} = \left(\langle \psi_k^{(\alpha\beta)}(\infty)\psi_j^{(\beta\beta)}(1)\psi_i^{(\beta\alpha)}(0)\rangle_{\mathbb{H}}\right)^N, \tag{2.37}$$

so we find a simple expression for these coefficients, in terms of mother BCFT boundary-boundary structure constants:

$$\boxed{\mathcal{B}_{\Psi_i,\Psi_j}^{(\beta\beta\alpha)\Psi_k} = \left(B_{ij}^{(\beta\beta\alpha)\,k}\right)^N.} \tag{2.38}$$

By similar considerations, the structure constants involving a non-diagonal BCCO $\Psi_{[i_1...i_N]}^{(\beta\alpha)}$ can be expressed as:

$$\boxed{\mathcal{B}_{\Psi_{[i_1...i_N]},\Psi_j}^{(\beta\beta\alpha)\Psi_k} = \sqrt{N}\prod_{a=1}^N B_{i_a j}^{(\beta\beta\alpha)\,k}} \tag{2.39}$$

The rest of the boundary-boundary structure constants of untwisted BCCOs can similarly be expressed in terms of mother BCFT quantities, but we will not need them in this work.

### 2.3.2 Orbifold bulk-boundary structure constants

The first bulk-boundary structure constant we need to calculate is $\mathcal{A}_{\sigma,\Psi_1}^{(\alpha)}$, where $\Psi_1^{(\alpha\alpha)}$ is just the identity boundary field. This can be expressed as the one-point function on the unit disk $\mathbb{D}$:

$$\mathcal{A}_{\sigma,\Psi_1}^{(\alpha)} = \langle \sigma(0,0)\rangle_{\mathbb{D}}^{\alpha}, \tag{2.40}$$

which is just the ratio of mother CFT partition functions:

$$\langle \sigma(0,0)\rangle_{\mathbb{D}}^{\alpha} = \frac{\mathcal{Z}_{\mathbb{D}_N}^{(\alpha)}}{\left[\mathcal{Z}_{\mathbb{D}}^{(\alpha)}\right]^N}, \tag{2.41}$$

where $\mathbb{D}_N$ denotes the $N$-th covering of the unit disk with branch points at 0 and 1. As shown in [76], we can express (2.40) in terms of the *ground state degeneracy* $g_\alpha = \langle 0|\alpha \rangle$ [69] (which is defined as the overlap between the vacuum state $|0\rangle$ and the boundary state $|\alpha\rangle$ in the mother BCFT):

$$\mathcal{A}^{(\alpha)}_{\sigma,\Psi_\mathbf{1}} = g_\alpha^{1-N} . \tag{2.42}$$

Using this result, we can calculate the one-point structure constants of composite twist operators $\sigma_i$, by using the definition (2.13) and the relation between twist correlators on the disk $\mathbb{D}$ and the mother CFT partition function on $\mathbb{D}_N$, which simply gives:

$$\mathcal{A}^{(\alpha)}_{\sigma_i,\Psi_\mathbf{1}} = \mathcal{A}^{(\alpha)}_{\sigma,\Psi_\mathbf{1}} A^\alpha_{\phi_i} , \tag{2.43}$$

where $A^\alpha_{\phi_i}$ is the mother CFT one-point structure constant of $\phi_i$ with conformal boundary condition $\alpha$. The proof is relegated to Appendix B.1.

Extending these results to more complicated bulk-boundary structure constants $\mathcal{A}^{(\alpha)}_{\sigma_i^{[k]},\Psi_j}$ for generic choices of mother CFT and cyclic group $\mathbb{Z}_N$ is not usually straightforward and depends on our knowledge of correlation functions in the mother CFT. For example, in Appendix B.2 we calculate the structure constant $\mathcal{A}^{(\alpha)}_{\sigma,\Psi_{13}}$ in the $\mathbb{Z}_2$ orbifold BCFT, since it can be expressed in terms of a two-point function of boundary operators in the mother CFT. For generic $N$ and composite twist operator $\sigma_i^{[k]}$, knowledge of higher-point correlators in the mother CFT is required to compute such structure constants through the same unfolding methods.

# 3 Differential equations in the $\mathbb{Z}_2$ and $\mathbb{Z}_3$ orbifold BCFT

## 3.1 Setup for the calculations

We consider the case of a generic BCFT, with central charge $c$. The model is defined on the upper half plane, with conformal boundary conditions $\alpha$ and $\beta$ set on the negative $\Re(z) < 0$ and positive $\Re(z) > 0$ parts of the real axis, respectively. We will work, for the entirety of this section, under the assumption that the most relevant BCCO interpolating between these boundary conditions is $\psi_{12}^{(\alpha\beta)}$, with conformal dimension $h_{12}$. This implies that the BCCO has a null-vector at level 2. Of course, our results also apply to the case where the BCCO is $\psi_{21}^{(\alpha\beta)}$, up to changing $g \to 1/g$.

In the $\mathbb{Z}_N$ orbifold of this theory, we will consider one-point correlators of generic *composite twist operators* $\sigma_i$ of twist charge $[k=1]$, in a background with mixed BC $\alpha$ and $\beta$, corresponding to the replicated boundary conditions of the mother BCFT. The change in boundary conditions in the orbifold theory will be implemented by the diagonal BCCO $\Psi_{12}^{(\alpha\beta)}$ defined in (2.27), with conformal dimension $h_{\Psi_{12}} = Nh_{12}$. Since we will aim to compare our CFT results with lattice data in Section 4, we will define our twist correlator on an infinite strip $\mathbb{S}$ of width $L$, parametrized by the complex coordinate $w = u + iv$, with $u \in [0, L]$ and $v \in \mathbb{R}$. The conformal boundary conditions on the $u = 0$ and $u = L$ sides of the strip are set to be $\alpha$ and $\beta$, respectively. We will consider correlators with a twist $\sigma_i$ inserted at $w = \ell$:

$$\langle \sigma_i(\ell,\ell) \rangle_\mathbb{S}^{\alpha\beta} , \tag{3.1}$$

where $\ell$ is measured from the boundary $\beta$, in accordance with Figure 1.

This correlator is now mapped to the upper half plane, through:

$$w = \frac{-iL}{\pi} \ln z \,, \tag{3.2}$$

and expressed, using (2.28), as:

$$\langle \sigma_i(z, \bar{z}) \rangle_{\mathbb{H}}^{\alpha\beta} = \langle \Psi_{12}^{(\alpha\beta)}(\infty) \, \sigma_i(z, \bar{z}) \Psi_{12}^{(\beta\alpha)}(0) \rangle_{\mathbb{H}} \,, \tag{3.3}$$

with $z = \exp i\pi\ell/L$ in terms of strip coordinates. Using the information about the operator algebra of the orbifold BCFT we have presented in Section 2.3, we can write the following block expansion for (3.1)

$$\langle \sigma_i(\ell, \ell) \rangle_{\mathbb{S}}^{\alpha\beta} = \mathcal{J} \sum_{\ell} \mathcal{A}_{\sigma_i, \Psi_\ell}^{(\beta)} \mathcal{B}_{\Psi_\ell, \Psi_{12}}^{(\beta\beta\alpha)\Psi_{12}} \widetilde{\mathcal{F}}_\ell(\eta) \,, \tag{3.4}$$

where $\eta = z/\bar{z} = \exp(2\pi i\ell/L)$, and $\mathcal{J} = (L\bar{z}/\pi)^{-2h_{\sigma_i}}$ is the combined Jacobian associated to the Möbius map $\zeta \mapsto \zeta/\bar{z}$ that takes $(0, z, \bar{z}, \infty) \mapsto (0, \eta, 1, \infty)$ and the map $w \mapsto z$ from the strip to the upper half plane. We recall that the $\widetilde{\mathcal{F}}_\ell$'s are the conformal blocks in the channel $\eta \to 1$.

As per Cardy's doubling argument [72], the functions $\tilde{\mathcal{F}}_k(\eta)$ are four-point conformal blocks (2.31) with $\Phi = \Phi_{12}$.

To proceed, we need to determine the differential equation satisfied by these functions. To this end, we will use a combination of the null-vector conditions and the orbifold Ward identities derived in the Appendix.

## 3.2 The function $\langle \Psi_{12} \cdot \sigma \cdot \Psi_{12} \rangle$ in a generic $\mathbb{Z}_2$ orbifold

Following the general approach described above, the function $\langle \Psi_{12}(\infty)\sigma(z, \bar{z})\Psi_{12}(0) \rangle_{\mathbb{H}}$ is given, up to the overall factor $\bar{z}^{-2h_\sigma}$, by a linear combination of the conformal blocks

$$\mathcal{F}_k(\eta) = \langle \Phi_{12} | \sigma(1) \mathcal{P}_k \sigma(\eta) | \Phi_{12} \rangle \,. \tag{3.5}$$

It turns out that this family of conformal blocks was already studied in [42], for the calculation of the single-interval Rényi entropy in the excited state $|\Phi_{12}\rangle$ with periodic BC. Let us recall how the derivation of the corresponding ODE goes.

We use the null-vectors at level two of the untwisted chiral state $|\Phi_{12}\rangle$, given by:

$$\begin{aligned} \left[ L_{-2}^{(0)} - \frac{1}{2g} \left( L_{-1}^{(0)} \right)^2 - \frac{1}{2g} \left( L_{-1}^{(1)} \right)^2 \right] \cdot |\Phi_{12}\rangle &\equiv 0, \\ \left[ L_{-2}^{(1)} - \frac{1}{g} L_{-1}^{(0)} L_{-1}^{(1)} \right] \cdot |\Phi_{12}\rangle &\equiv 0 \,, \end{aligned} \tag{3.6}$$

and the null vector at level $1/N$ of the bare twist operator $\sigma$:

$$L_{-1/2}^{(1)} \cdot \sigma \equiv 0 \,. \tag{3.7}$$

We combine these with the orbifold Ward identity for the *chiral* correlator:

$$\mathcal{G}^{(1)}(w, \eta) = \langle \Phi_{12} | \sigma(1) \mathcal{P}_k \sigma(\eta) T^{(1)}(w) L_{-1}^{(1)} | \Phi_{12} \rangle \,, \tag{3.8}$$

with $(m_1, m_2, m_3, m_4) = (0, -1/2, -1/2, -1)$ in the notation of (C.1). This gives, after taking into account (3.7):

$$\sum_{p=0,1,2} d_p \langle \Phi_{12} | \sigma(1) \mathcal{P}_k \sigma(\eta) L^{(1)}_{-p+2} L^{(1)}_{-1} | \Phi_{12} \rangle = 0 , \tag{3.9}$$

with $d_p$ calculated from the series $(C.6)$.

By substituting the null vectors (2.18–3.7) and employing the identity (C.7), one obtains the differential equation:

$$\begin{aligned} 64g^2\eta^2(\eta-1)^2\, \partial_\eta^2\mathcal{F} + 16g\eta(\eta-1)\left[(-14g^2+23g-6)\eta+2g(1-4g)\right]\partial_\eta\mathcal{F} \\ + (3g-2)\left[+3(5g-6)(1-2g)^2\eta^2+12g(1-2g)\eta+16g^2(g-1)\right]\mathcal{F} = 0 , \end{aligned} \tag{3.10}$$

whose Riemann scheme is given by:

| 0 | 1 | $\infty$ |
|---|---|---|
| $-2h_{12}$ | $-2h_\sigma$ | $2h_\sigma - 2h_{12}$ |
| $-2h_{12} + h_{13}/2$ | $-2h_\sigma + 2h_{13}$ | $2h_\sigma - 2h_{12} + h_{13}/2$ |

This corresponds to the intermediary states $\{\sigma, \sigma_{13}\}$ in the channels $\eta \to 0$ and $\eta \to \infty$, and $\{\mathbf{1}, \Phi_{13}\}$ in the channel $\eta \to 1$. Note that, when the mother CFT is a minimal model $\mathcal{M}_{p,p'}$, one can check for various values of $(p, p')$ that these are exactly the intermediary states allowed by the orbifold fusion given in F, and that they all have multiplicity one.

To proceed, one can define the shifted function $f(\eta)$:

$$f(\eta) = (1-\eta)^{2h_\sigma}\eta^{2h_{12}}\mathcal{F}(\eta) , \tag{3.11}$$

and substitute in (3.10) to find that $f(\eta)$ satisfies a second order hypergeometric equation (E.1) with parameters:

$$a = 2 - 3g , \qquad b = \frac{3}{2} - 2g , \qquad c = \frac{3}{2} - g . \tag{3.12}$$

We can work with the basis (E.3) of solutions around $\eta = 1$ for this hypergeometric equation— in the block expansion (3.4), this corresponds to approaching the twist operator to the boundary $\beta$. Thus, the conformal blocks $\widetilde{\mathcal{F}}_\ell(\eta)$ we seek are:

$$\begin{aligned} \widetilde{\mathcal{F}}_{\mathbf{1}}(\eta) &= (1-\eta)^{-2h_\sigma}\eta^{-2h_{12}}{}_2\mathrm{F}_1(a, b; a+b-c+1 \mid 1-\eta) , \\ \widetilde{\mathcal{F}}_{13}(\eta) &= (1-\eta)^{-2h_\sigma+2h_{13}}\eta^{-2h_{12}}{}_2\mathrm{F}_1(c-b, c-a; c-a-b+1 \mid 1-\eta) , \end{aligned} \tag{3.13}$$

and they are normalized to be of the form $\widetilde{\mathcal{F}}_\ell(\eta) \sim (1-\eta)^{-2h_\sigma+h}(1+\dots)$ as $\eta \to 1$ , where $h$ is the conformal dimension of the internal operator of the conformal block in this channel. The bulk-boundary fusion rules corresponding to the exponents are

$$\sigma\Big|_\beta \to \Psi_{\mathbf{1}} + \Psi_{13} , \tag{3.14}$$

as $\sigma$ is approached to the conformal boundary $\beta$. Substituting the blocks and the expressions for the orbifold BCFT structure constants in (3.4) one finds the expression of the one-point twist correlator

$$\langle \sigma(z, \bar{z}) \rangle_{\mathbb{H}}^{\alpha\beta} = \bar{z}^{-2h_\sigma}\left[\mathcal{A}^{(\beta)}_{\sigma,\Psi_{\mathbf{1}}}\mathcal{B}^{(\beta\beta\alpha)\Psi_{12}}_{\Psi_{\mathbf{1}},\Psi_{12}}\tilde{\mathcal{F}}_{\mathbf{1}}(\eta) + \mathcal{A}^{(\beta)}_{\sigma,\Psi_{13}}\mathcal{B}^{(\beta\beta\alpha)\Psi_{12}}_{\Psi_{13},\Psi_{12}}\tilde{\mathcal{F}}_{13}(\eta)\right] , \tag{3.15}$$

where the various structure constants are expressed in terms of the mother BCFT data as

$$\mathcal{A}^{(\beta)}_{\sigma,\Psi_1} = \mathcal{A}^{(\beta)}_{\sigma,\Psi_{13}} = g_\beta^{-1}, \qquad \mathcal{B}^{(\beta\beta\alpha)\Psi_{12}}_{\Psi_1,\Psi_{12}} = 1, \qquad \mathcal{B}^{(\beta\beta\alpha)\Psi_{12}}_{\Psi_{13},\Psi_{12}} = \left( B^{(\beta\beta\alpha)\,\psi_{12}}_{\psi_{13}\psi_{12}} \right)^2. \tag{3.16}$$

It is interesting now to observe that, for some pairs of conformal BCs $(\alpha, \beta)$, the block expansion (3.15) greatly simplifies because the boundary-boundary structure constant $\mathcal{B}^{(\beta\beta\alpha)\Psi_{12}}_{\Psi_{13},\Psi_{12}}$ vanishes. At the level of the mother BCFT, this is equivalent to demanding that the operator $\psi^{(\beta\beta)}_{13}$ is not allowed in the theory.

For BCFTs based on $A$-series minimal models $\mathcal{M}(p, p')$, this holds for any pair of mixed conformal BCs $(\alpha, \beta) \equiv (\phi_{r2}, \phi_{r1})$, labelled by bulk primary fields with $1 \le r < p$. One can use well-established results about fusion rules in such models [77], to check that:

$$\phi_{12} \in \phi_{r1} \times \phi_{r2} \tag{3.17}$$

so that these BCs are interpolated by $\psi^{(\alpha\beta)}_{12}$ and

$$\phi_{13} \notin \phi_{r1} \times \phi_{r1} \tag{3.18}$$

which implies that $\psi^{(\beta\beta)}_{13}$ is not in the boundary operator spectrum of the BCFT.

Under these conditions, the correlator in (3.15) simplifies to:

$$\langle \sigma(z, \bar{z}) \rangle^{\alpha\beta}_{\mathbb{H}} = \eta^{h_\sigma} \left[ g_{\phi_{r1}}^{-1} \tilde{\mathcal{F}}_1(\eta) \right], \tag{3.19}$$

so that one finds the second Rényi entropy in the strip setup to be:

$$S^{(\alpha,\beta)}_2(\ell) \sim \frac{c}{8} \log \frac{2L}{\pi a} \sin\left( \frac{\pi\ell}{L} \right) + \log g_{\phi_{r1}} - \log \left[ \eta^{-2h_{12}} {}_2F_1\left( 2 - 3g, 3/2 - g; 3 - 4g \mid 1 - \eta \right) \right] \tag{3.20}$$

It is now interesting to check that the theoretical prediction for this kind of mixed BC has the expected behaviour near the $\phi_{r1}$ and $\phi_{r2}$ boundaries. It is known [2] that the second Rényi entropy of an interval $\ell$ touching one of the *identical* boundaries of a finite system of size $L$ is given by:

$$S^{(\alpha,\alpha)}_2([0, \ell]) = \frac{c}{8} \log \left[ \frac{2L}{\pi} \sin\left( \frac{\pi\ell}{L} \right) \right] + \log g_\alpha \tag{3.21}$$

We then anticipate that the mixed BC result (3.20) will interpolate between $S^{(\phi_{r1},\phi_{r1})}_2$ and $S^{(\phi_{r2},\phi_{r2})}_2$ as $\ell \to 0$ and $\ell \to L$

Furthermore, suppose we consider the *difference* in second Rényi entropies between two mixed BC setups $(\phi_{r2}, \phi_{r1})$ and $(\phi_{r'2}, \phi_{r'1})$ for the same bulk CFT. We then find the following universal result:

$$\boxed{\Delta S_2 = S^{(\phi_{r'2},\phi_{r'1})}_2 - S^{(\phi_{r2},\phi_{r1})}_2 = \log \frac{g_{\phi_{r1}}}{g_{\phi_{r'1}}} = \log \frac{g_{\phi_{r2}}}{g_{\phi_{r'2}}}} \tag{3.22}$$

where the latter equation follows from the expression of $g_{\phi_{r,s}} = S_{\phi_{r,s},\phi_0}/\sqrt{S_{\mathbf{1},\phi_0}}$ [69] in terms of S-matrix elements of minimal models [77] - here $\phi_0$ denotes the field with the lowest conformal dimension of the *diagonal* bulk CFT.

This entropy difference is thus determined, in these cases, by the identical BC expressions (3.21) that describe the asymptotic behaviour near the boundaries of the mixed BC setup. We illustrate these ideas graphically for the $A$-series minimal model $\mathcal{M}(6, 5)$ in Figure 2, by plotting the second Rényi entropies shifted by $L^{2h_\sigma}$ for two mixed BC setups $(\phi_{12}, \phi_{11})$ and $(\phi_{32}, \phi_{31})$ for the BCFTs based on $\mathcal{M}(6, 5)$, together with the relevant identical BC setups for each case.

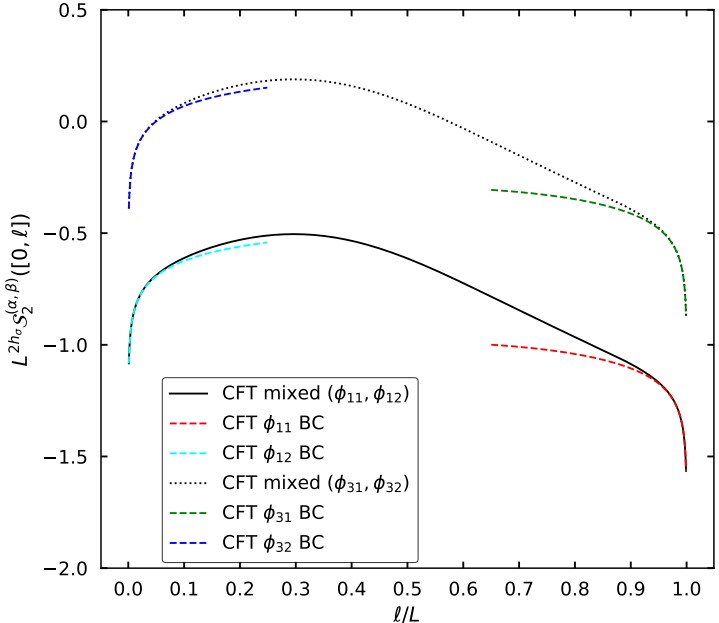

Figure 2: Shifted second Rényi entropies for the mixed BC setups $(\phi_{12}, \phi_{11})$ and $(\phi_{32}, \phi_{31})$ for the BCFTs based on $\mathcal{M}(6,5)$. The difference between the two curves is constant with respect to the interval size, and thusly fixed by their respective asymptotic behaviours around $\ell \to 0$ and $\ell \to L$

## 3.3 The function $\langle \Psi_{12} \cdot \sigma_h \cdot \Psi_{12} \rangle$ in a generic $\mathbb{Z}_2$ orbifold

From the perspective of critical quantum chains, the result in (3.15) only determines the leading contribution to the second Rényi entropy. To understand finite-size corrections to this result, we should also study the one-point function of subleading primary twist operators, namely $\langle \sigma_h(z, \bar{z}) \rangle_{\mathbb{H}}^{\alpha\beta}$.

We shall derive an ODE for the conformal blocks

$$\mathcal{F}_k(\eta) = \langle \Phi_{12} | \sigma_h(1) \mathcal{P}_k \sigma_h(\eta) | \Phi_{12} \rangle \,. \tag{3.23}$$

Here we consider the case of a generic composite twist operator $\sigma_h$ with conformal dimension

$$\widehat{h} = h_\sigma + \frac{h}{N} \,, \tag{3.24}$$

and thus we do not assume any null-vector condition on $\sigma_h$. Besides the null-vector conditions at level two (3.6), we will need the null-vector at level three in the module of $|\Phi_{12}\rangle$:

$$L_{-2}^{(1)} L_{-1}^{(1)} |\Phi_{12}\rangle = \left[ -L_{-3}^{(0)} + \frac{1}{g} \left( 2g L_{-1}^{(0)} L_{-2}^{(0)} - \left( L_{-1}^{(0)} \right)^3 \right) \right] |\Phi_{12}\rangle \,, \tag{3.25}$$

and two Ward identities obtained from:

$$\langle \Phi_{12} | \sigma_h(1) \mathcal{P}_k \sigma_h(\eta) T^{(1)}(z) L_{-1}^{(1)} |\Phi_{12}\rangle \,, \tag{3.26}$$

with $(m_1, m_2, m_3, m_4) = (-1, 1/2, 1/2, -2)$:

$$a_{0|1} \langle \Phi_{12} | L_1^{(1)} \sigma_h(1) \mathcal{P}_k \sigma_h(\eta) L_{-1}^{(1)} |\Phi_{12}\rangle = \sum_{p=0}^{3} d_{p|1} \langle \Phi_{12} | \sigma_h(1) \mathcal{P}_k \sigma_h(\eta) L_{-2+p}^{(1)} L_{-1}^{(1)} |\Phi_{12}\rangle \tag{3.27}$$

| 0 | 1 | $\infty$ |
|---|---|---|
| $-2h_{12}$ | $-2\widehat{h}_\alpha$ | $2\widehat{h}_\alpha - 2h_{12}$ |
| $-2h_{12} + \frac{1}{2}$ | $-2\widehat{h}_\alpha + h_{13}$ | $2\widehat{h}_\alpha - 2h_{12} + \frac{1}{2}$ |
| $-\widehat{h}_\alpha - 2h_{12} + \widehat{h}_{\alpha+b}$ | $-2\widehat{h}_\alpha + 2h_{13}$ | $\widehat{h}_\alpha - 2h_{12} + \widehat{h}_{\alpha+b}$ |
| $-\widehat{h}_\alpha - 2h_{12} + \widehat{h}_{\alpha-b}$ | $-2\widehat{h}_\alpha + 2h_{13} + 2$ | $\widehat{h}_\alpha - 2h_{12} + \widehat{h}_{\alpha-b}$ |

Table 1: Singular exponents around $\eta = 0, 1, \infty$

and $(m_1, m_2, m_3, m_4) = (-2, 1/2, 1/2, -1)$ :

$$a_{0|2} \langle \Phi_{12}| L_2^{(1)} \sigma_h(1) \mathcal{P}_k \sigma_h(x) L_{-1}^{(1)} |\Phi_{12}\rangle + a_{1|2} \langle \Phi_{12}| L_1^{(1)} \sigma_h(1) \mathcal{P}_k \sigma_h(x) L_{-1}^{(1)} |\Phi_{12}\rangle =$$
$$= d_{0|2} \langle \Phi_{12}| \sigma_h(1) \mathcal{P}_k \sigma_h(\eta) L_{-1}^{(1)} L_{-1}^{(1)} |\Phi_{12}\rangle + d_{1|2} \langle \Phi_{12}| \sigma_h(1) \mathcal{P}_k \sigma_h(\eta) L_0^{(1)} L_{-1}^{(1)} |\Phi_{12}\rangle +$$
$$+ d_{2|2} \langle \Phi_{12}| \sigma_h(1) \mathcal{P}_k \sigma_h(\eta) L_1^{(1)} L_{-1}^{(1)} |\Phi_{12}\rangle \quad (3.28)$$

Putting everything together, and applying the change of function

$$\mathcal{F}(\eta) = \eta^{-2h_{12}} (1 - \eta)^{4h_{12} - 2\widehat{h}} f(\eta) , \quad (3.29)$$

we obtain the fourth-order ODE

$$(\eta - 1)^4 \eta^3 \partial_\eta^4 f + \frac{1}{2}(\eta - 1)^3 \eta^2 \left[ (2g + 13)\eta + (2g - 11) \right] \partial_\eta^3 f$$

$$- \frac{1}{8}(\eta - 1)^2 \eta \left[ (16g\widehat{h} + 6g^2 - 45g - 60)\eta^2 + (20g^2 + 34g + 96)\eta + (16g\widehat{h} + 6g^2 + 3g - 36) \right] \partial_\eta^2 f$$

$$- \frac{g}{16}(\eta - 1) \left[ (48\widehat{h} + 18g - 75)\eta^3 + (16g^2 - 18g - 112\widehat{h} + 167)\eta^2 \right.$$

$$\left. + (80\widehat{h} + 16g^2 - 74g - 53)\eta + (-16\widehat{h} - 6g + 9) \right] \partial_\eta f$$

$$+ \frac{g}{8} \left[ (16g^2\widehat{h} + 6g^3 - 13g^2 + 4)\eta^2 + (-32g^2\widehat{h} + 12g^3 + 34g^2 - 64g + 24)\eta \right.$$

$$\left. + (24 + 16g^2\widehat{h} + 6g^3 - 13g^2 + 4) \right] f = 0 . \quad (3.30)$$

At this stage, it will be convenient to use the Coulomb-Gas parametrization to analyse the local exponents of the ODE. Recall the relation between the mother CFT central charge and the parameter $g$:

$$c = 1 - 24Q^2 , \qquad Q = \frac{1}{2}(1/b - b) , \qquad b = \sqrt{g} . \quad (3.31)$$

The conformal dimensions in the mother CFT are parametrized by the *vertex charge* $\alpha$ as

$$h_\alpha = \alpha(\alpha - 2Q) , \quad (3.32)$$

and we use the shorthand notation for the conformal dimension of composite twisted operators

$$\widehat{h}_\alpha = h_\sigma + \frac{h_\alpha}{N} . \quad (3.33)$$

In this parametrization, the Riemann scheme for $\mathcal{F}(\eta)$ is given in Table 1.

These exponents correspond to the intermediary states (counted with their multiplicities):

$$\{\mathbf{1}, [\mathbf{1}, \phi_{13}], \Phi_{13}, \Phi_{13}\} \qquad \text{in the channel } \eta \to 1 \,,$$
$$\{\sigma_h, L^{(1)}_{-1/2} \cdot \sigma_h, \sigma_{h'}, \sigma_{h''}\} \quad \text{in the channels } \eta \to 0 \text{ and } \eta \to \infty \,. \tag{3.34}$$

Here, we have defined $h' = h_{\alpha+b}$ and $h'' = h_{\alpha-b}$. Recall that the conformal blocks are labelled by primary operators under the neutral subalgebra $A$, and that $L^{(1)}_{-1/2} \cdot \sigma_h$ is one of these operators. When the mother CFT is a minimal model, one can check on various examples that the orbifold fusion rules derived in [70] from the Verlinde formula are consistent with these intermediary states.

While an analytic solution to the differential equation is not known, one can determine the conformal blocks $\mathcal{F}_k(\eta)$ around $\eta = 0$ and $\widetilde{\mathcal{F}}(\eta)$ around $\eta = 1$ numerically to arbitrary precision. Assuming this step has been performed, all that is left is to calculate the structure constants in the block expansion (3.4). The boundary-boundary structure constants are calculated from (2.38) and (2.39), while the bulk-boundary structure constants can be calculated analytically through unfolding, as shown, for some cases in Appendix B.2 and B.1. For numerical studies, however, it is simpler to bootstrap some of the coefficients in the block expansion (3.4) rather than to calculate all of them analytically.

To implement this method, as detailed in [78], one needs to compare the block expansions in (3.4) with the block expansion corresponding to sending the twist field to the part of the boundary, endowed with the $\alpha$ BC. The crucial point here is that the conformal blocks $\mathcal{F}_k(\eta)$ are branched functions on $\mathbb{C}$, with branch points $\{0, 1, \infty\}$, and, thus, sending the twist field to the boundary with BC $\alpha$ is equivalent to crossing to the other branch of the function. This is marked by appending the phase factor $e^{2\pi i}$ to the variable $\eta$ to get:

$$\langle \sigma_h(z, \bar{z}) \rangle^{\alpha\beta}_{\mathbb{S}} = \mathcal{J}_{SL(2,\mathbb{C})} \sum_\ell \mathcal{A}^{(\alpha)}_{\sigma_h, \Psi_\ell} \mathcal{B}^{(\alpha\alpha\beta)\Psi_{12}}_{\Psi_\ell, \Psi_{12}} \widetilde{\mathcal{F}}_\ell(e^{2\pi i}\eta) \,. \tag{3.35}$$

To proceed, one needs to find the *monodromy matrix $X$* around zero for the basis $\widetilde{\mathcal{F}}_\ell(\eta)$, which encodes the behaviour of the conformal blocks as the branch cut is crossed:

$$\widetilde{\mathcal{F}}_\ell(e^{2\pi i}\eta) = \sum_m X_{\ell m} \widetilde{\mathcal{F}}_m(\eta) \,. \tag{3.36}$$

Since the monodromy of the blocks $\widetilde{\mathcal{F}}_\ell$ around zero is non-diagonal, we can use the fusing matrix $P_{ij}$ to express the blocks $\widetilde{\mathcal{F}}_\ell$ in terms of a basis of the blocks $\mathcal{F}_k$, which have diagonal monodromy around $\eta = 0$:

$$\widetilde{\mathcal{F}}_\ell(\eta) = \sum_k P_{\ell k} \mathcal{F}_k(\eta) \,. \tag{3.37}$$

The blocks $\mathcal{F}_k(\eta)$ simply acquire a phase under $z \to e^{2\pi i}z$, so their monodromy matrix $Y$ is diagonal:

$$\mathcal{F}_k(e^{2i\pi}\eta) = \sum_j Y_{kj}\, \mathcal{F}_j(\eta)\,, \qquad Y_{kj} = \delta_{kj} \exp\left[2\pi i \left(-\widehat{h}_\alpha - 2h_{12} + \widehat{h}_k\right)\right]\,, \tag{3.38}$$

where the exponents in the exponential above, are simply read off from the Riemann scheme. Then, the monodromy matrix of the blocks $\widetilde{\mathcal{F}}_\ell(\eta)$ is found from the matrix product:

$$X = P \cdot Y \cdot P^{-1}\,, \tag{3.39}$$

which allows us to compare the block expansions in (3.4) and (3.35) to find a duality relation, of the type presented in [78]:

$$\mathcal{A}^{(\beta)}_{\sigma_h,\Psi_i} \mathcal{B}^{(\beta\beta\alpha)\Psi_i}_{\Psi_{12},\Psi_{12}} = \sum_j \mathcal{A}^{(\alpha)}_{\sigma_h,\Psi_j} \mathcal{B}^{(\alpha\alpha\beta)\Psi_j}_{\Psi_{12},\Psi_{12}} X_{ji}\,. \tag{3.40}$$

Using the numerical determinations for $\mathcal{F}_k(\eta)$ and $\widetilde{\mathcal{F}}_\ell(\eta)$ , one can find a good estimate for the fusing matrix $P_{ij}$, and, consequently, $X_{ij}$. A more fleshed out example of how the determination of $P_{ij}$ works has been relegated to the Appendix H, where this equation is used for the case of the $\mathbb{Z}_2$ orbifold of the three-state Potts model BCFT.

After solving the linear system in (3.40) one can evaluate the unknown structure constants $\mathcal{A}^{(\beta)}_{\sigma_h,\Psi_i}$ and $\mathcal{A}^{(\alpha)}_{\sigma_h,\Psi_i}$. At this point, we stress that (3.40) gives, at most, *four* constraints between the unknown structure constants. To fully determine all these quantities, one should calculate the remaining four structure constants through other methods.

## 3.4 The function $\langle \Psi_{12} \cdot \sigma \cdot \Psi_{12} \rangle$ in a generic $\mathbb{Z}_3$ orbifold

Here the relevant conformal blocks are

$$\mathcal{F}_k(\eta) = \langle \Phi_{12} | \sigma(1) \mathcal{P}_k \sigma^\dagger(\eta) | \Phi_{12} \rangle\,. \tag{3.41}$$

We give the null-vectors of $|\Phi_{12}\rangle$ at levels two and three:

$$L^{(r)}_{-2} |\Phi_{12}\rangle = \frac{1}{3g} \sum_{s=0}^{2} L^{(r-s)}_{-1} L^{(s)}_{-1} |\Phi_{12}\rangle\,, \tag{3.42}$$

$$L^{(3-r)}_{-1} L^{(r)}_{-2} |\Phi_{12}\rangle = \frac{1}{3g} \left[ 2L^{(0)}_{-1} L^{(1)}_{-1} L^{(2)}_{-1} + \left( L^{(3-r)}_{-1} \right)^3 \right] |\Phi_{12}\rangle\,, \tag{3.43}$$

for $r \in \{0,1,2\}$. We will also need the null-vectors for the out-state $\langle \Phi_{12}|$, which can be obtained by Hermitian conjugation (2.6).

To derive an ODE for the conformal blocks, we had to employ seven orbifold Ward identities, together with six of the null-vector conditions above. To not overload the presentation of this section with technical details, we relegate the specifics of the derivation to Appendix G. We apply the change of function

$$\mathcal{F}(\eta) = \eta^{-8h_{12}/3} (1-\eta)^{16h_{12}/3 - 2h_\sigma} f(\eta)\,. \tag{3.44}$$

The function $f$ satisfies the ODE

$$(\eta - 1)^3 \eta^2\, \partial_\eta^3 f + (\eta-1)^2 \eta\, [(g+3)\eta + (g-3)]\, \partial_\eta^2 f$$
$$+ \frac{2}{9}(\eta-1) \left[ 2(2+3g)\eta^2 - 2(7 - 15g + 18g^2)\eta + (4-3g) \right] \partial_\eta f \tag{3.45}$$
$$+ \frac{4}{27}(1-6g)(2-3g)(\eta+1)\, f = 0\,.$$

The Riemann scheme for $\mathcal{F}$ is

| $0$ | $1$ | $\infty$ |
|---|---|---|
| $-\frac{8}{3}h_{12}$ | $-2h_\sigma$ | $2h_\sigma - \frac{8}{3}h_{12}$ |
| $-\frac{8}{3}h_{12} + \frac{1}{3}$ | $-2h_\sigma + 2h_{13}$ | $2h_\sigma - \frac{8}{3}h_{12} + \frac{1}{3}$ |
| $-3h_{12} + \frac{h_{14}}{3}$ | $-2h_\sigma + 3h_{13}$ | $2h_\sigma - 3h_{12} + \frac{h_{14}}{3}$ |

The local exponents correspond to the intermediary states:

$$
\begin{aligned}
\{\mathbf{1}, [\mathbf{1}, \phi_{13}, \phi_{13}], \Phi_{13}\} \quad &\text{in the channel } \eta \to 1\,, \\
\{\sigma_{12}, L^{(1)}_{-1/3} \cdot \sigma_{12}, \sigma_{14}\} \quad &\text{in the channels } \eta \to 0 \text{ and } \eta \to \infty\,.
\end{aligned}
\tag{3.46}
$$

In the orbifold BCFT, this translates into the following fusion rules for the twist operator with the boundary $\beta$:

$$
\sigma_{\mathbf{1}}\Big|_{\beta} \to \Psi^{(\beta\beta)}_{\mathbf{1}} + \Psi^{(\beta\beta)}_{[\mathbf{1},\phi_{13},\phi_{13}]} + \Psi^{(\beta\beta)}_{13}\,.
\tag{3.47}
$$

The analytic solutions to the differential equation (3.45) are not known, but they can be evaluated numerically, to arbitrary precision. Then, one can use the bootstrap to determine some relations between the unknown structure constants in the expansion (3.4), as outlined in the previous section, and determine the rest analytically, by unfolding methods, to complete the calculation of the mixed BC correlator of the bare twist.

We note that a fourth order differential equation that the correlator (2.28) satisfies has already been found in [75], where it plays a role in the determination of the leading contribution to the third Rényi entropy of an excited state in a periodic 1D critical chain. As predicted in [75], there is no degeneracy in the exponents in the more constraining third order differential equation we have found here. Note that these exponents are the ones expected from the orbifold fusion rules [70].

## 3.5 The function $\langle \Psi_{12} \cdot \sigma_{13} \cdot \Psi_{12} \rangle$ in the $\mathbb{Z}_3$ orbifold of the Ising model

In this section, we will work with the $\mathbb{Z}_3$ orbifold of the Ising BCFT. The bulk primary fields of this BCFT are $\phi_{11} \equiv \mathbf{1}$, $\phi_{12} \equiv s$ and $\phi_{13} \equiv \varepsilon$ with $h_s = 1/16$ and $h_\varepsilon = 1/2$. We will keep labelling the fields by their Kac indices, to not overcomplicate the notation.

We will provide here an alternative method for finding a differential equation for the one-point function:

$$
\langle \sigma_{13}(z, \bar{z}) \rangle^{f+}_{\mathbb{H}}\,,
\tag{3.48}
$$

where the orbifold conformal boundary conditions $\alpha = f$ and $\beta = +$ correspond to setting fixed and free BC respectively, on all the copies of the Ising mother BCFT. The diagonal BCCO $\Psi^{(f+)}_{12}$ is the one interpolating between them in the orbifold, since in the Ising BCFT only the $\psi^{(f+)}_{12}$ primary boundary field can change between the CBCs $(+) \leftrightarrow (f)$ [78]. As in the previous sections, we aim to find a differential equation satisfied by the conformal blocks

$$
\mathcal{F}_k(\eta) = \langle \Phi_{12} | \sigma_{13}(1) \mathcal{P}_k \sigma^{\dagger}_{13}(\eta) | \Phi_{12} \rangle\,.
\tag{3.49}
$$

First, we use the fusion numbers in (F.1) to infer the dimension of the space of conformal blocks for (3.48):

$$
\sum_i \mathcal{N}^i_{\sigma_{13},\sigma^{\dagger}_{13}} \mathcal{N}^{\Phi_{12}}_{i,\Phi_{12}} = 2\,,
\tag{3.50}
$$

which means the differential equation we seek should be second order.

By using the null-vectors induced on $\sigma_{13}$ together with the right combination of Ward identities, one should be able to rigorously derive it. Instead, we will assume this equation exists and is of Fuchsian type – a linear homogenous ODE whose three singular points are

regular. The latter assumption is based on the observation that the method exploited in the previous sections relies on expressing orbifold modes $L_m^{(r\neq 0)}$ in terms of Virasoro generators, whose combined differential action on correlators is well-understood in the literature [77], [79] to be of Fuchsian type.

Now, using the fusion numbers of (F.1), the fusion rules are

$$\sigma_{13} \times \Phi_{12} \to \sigma_{12} + L_{-2/3}^{(2)} \cdot \sigma_{12}\,,$$
$$\sigma_{13} \times \sigma_{13}^{\dagger} \to \mathbf{1} + [\mathbf{1}, \phi_{13}, \phi_{13}]\,,$$

(3.51)

so we can determine the asymptotic behaviour of the solutions around the regular singular points $\eta \in \{0, 1, \infty\}$ of the differential equation and infer the Riemann scheme:

| $0$ | $1$ | $\infty$ |
|---|---|---|
| $-h_{\sigma_{13}} - 3h_{12} + h_{\sigma_{12}}$ | $-2h_{\sigma_{13}}$ | $h_{\sigma_{13}} - 3h_{12} + h_{\sigma_{12}}$ |
| $-h_{\sigma_{13}} - 3h_{12} + h_{\sigma_{12}} + \frac{2}{3}$ | $-2h_{\sigma_{13}} + 2h_{13}$ | $h_{\sigma_{13}} - 3h_{12} + h_{\sigma_{12}} + \frac{2}{3}$ |

One can readily check that the entries of this Riemann scheme sum up to one, so, by virtue of a general theorem on Fuchsian ODEs (see [80]), there is a unique second-order Fuchsian ODE with this set of singular exponents. If we define the shifted function $f(\eta)$:

$$f(\eta) = \eta^{h_{\sigma_{13}} + 3h_{12} - h_{\sigma_{12}}} (1 - \eta)^{2h_{\sigma_{13}}} \mathcal{F}(\eta)\,,$$

(3.52)

we find, by the same considerations, that it should satisfy a second-order Fuchsian differential equation with the Riemann scheme

| $0$ | $1$ | $\infty$ |
|---|---|---|
| $0$ | $0$ | $-2h_{\sigma_{13}} - 6h_{12} + 2h_{\sigma_{12}}$ |
| $\frac{2}{3}$ | $2h_{13}$ | $-2h_{\sigma_{13}} - 6h_{12} + 2h_{\sigma_{12}} + \frac{2}{3}$ |

This is just the canonical Riemann scheme of a hypergeometric differential equation (E.1), with coefficients:

$$a = -2h_{\sigma_{13}} - 6h_{12} + 2h_{\sigma_{12}} = -2/3\,,$$
$$b = -2h_{\sigma_{13}} - 6h_{12} + 2h_{\sigma_{12}} + \frac{2}{3} = 0\,,$$
$$c = \frac{1}{3}\,,$$

(3.53)

in the conventions of Appendix E. We notice that the exponents in the $\eta \to 1$ channel are spaced by one, so we will have to deal with the *degenerate exponents* to arrive at a closed form solution. To do this, we will use the basis of solutions in the $\eta \to 0$ channel – given in (E.2) – to construct a linearly independent basis of solutions around $\eta \to 1$. The solutions for $f$ can be simplified, in this case, to:

$$I_1(\eta) = 1\,, \qquad I_2(\eta) = \eta^{2/3}\,,$$

(3.54)

which gives the conformal blocks around $\eta \to 0$:

$$\mathcal{F}_1(\eta) = \eta^{-1/3}(1 - \eta)^{-4/9}\,, \qquad \mathcal{F}_2(\eta) = \eta^{1/3}(1 - \eta)^{-4/9}\,.$$

(3.55)

in our normalisation convention.

To build the basis of solutions around $\eta = 1$, we look for the linear combinations $\tilde{\mathcal{F}}_i(\eta) = \sum_j P_{ij}^{-1} \mathcal{F}_j(\eta)$ that have the following series expansion around $\eta = 1$ :

$$\tilde{\mathcal{F}}_1(\eta) = (1-\eta)^{-4/9}\left(1 + \mathcal{O}[(1-\eta)^2]\right), \qquad \tilde{\mathcal{F}}_2(\eta) \sim (1-\eta)^{5/9}, \qquad (3.56)$$

since the power series associated to the orbifold identity should have no $(1-\eta)$ term due to the null-vectors $L_{-1}^{(r)} \cdot \mathbf{1} \equiv 0$, and the both solutions should have the leading coefficient normalised to one, in our convention for the conformal blocks. With these requirements, one finds the fusing matrix $P_{ij}^{-1}$ to be:

$$P^{-1} = \frac{1}{2}\begin{pmatrix} 1 & 1 \\ 3 & -3 \end{pmatrix}. \qquad (3.57)$$

Thus, the conformal blocks of (3.48) around $\eta = 1$ are found to be:

$$\tilde{\mathcal{F}}_1(\eta) = \frac{\eta^{-1/3} + \eta^{1/3}}{2(1-\eta)^{4/9}}, \qquad \tilde{\mathcal{F}}_2(\eta) = \frac{3(\eta^{-1/3} - \eta^{1/3})}{2(1-\eta)^{4/9}}. \qquad (3.58)$$

For the physical correlation function, we write

$$\langle \sigma_{13}(z,\bar{z})\rangle_{\mathbb{H}}^{f+} = \bar{z}^{-2h_{\sigma_{1,3}}}\left[\mathcal{A}_{\sigma_{13},\Psi_{\mathbf{1}}}^{(+)}\mathcal{B}_{\Psi_{\mathbf{1}},\Psi_{12}}^{(++f)\Psi_{12}}\tilde{\mathcal{F}}_1(\eta) + \mathcal{A}_{\sigma_{13},[\psi_{\mathbf{1}},\psi_{13},\psi_{13}]}^{(+)}\mathcal{B}_{[\psi_{\mathbf{1}},\psi_{13},\psi_{13}],\Psi_{12}}^{(++f)\Psi_{12}}\tilde{\mathcal{F}}_2(\eta)\right]. \qquad (3.59)$$

Finally, we observe that $B_{\psi_{13}\psi_{12}}^{(++f)\psi_{12}}$ vanishes, and hence $\mathcal{B}_{[\psi_{\mathbf{1}},\psi_{13},\psi_{13}],\Psi_{12}}^{(++f)\Psi_{12}} = 0$, so the expression (3.59) simplifies to:

$$\boxed{\langle \sigma_{13}(z,\bar{z})\rangle_{\mathbb{H}}^{f+} = 2^{5/9} \times \frac{\cos(2\theta/3)}{(r\sin\theta)^{4/9}}, \qquad z = re^{i\theta}} \qquad (3.60)$$

where we have also used:

$$\mathcal{A}_{\sigma_{13},\Psi_{\mathbf{1}}}^{(+)} = g_+^{-2}, \qquad \mathcal{B}_{\Psi_{\mathbf{1}},\Psi_{12}}^{(++f)\Psi_{12}} = 1, \qquad (3.61)$$

and the value of the ground-state degeneracy for fixed BC $g_+ = 1/\sqrt{2}$ in the Ising BCFT [69].

## 3.6 More hypergeometric differential equations in the Ising cyclic orbifold BCFTs

We have managed, in Sections 3.3 and 3.4 to obtain differential equations for cyclic orbifolds of generic mother BCFTs, but have not been able to provide analytic solutions for them.

One can, however, find second order differential equations for particular choices of $\mathcal{M}_{p,p'}$ and composite twist fields (for the correlators of Section 3.3), in the manner presented in Section 3.5, which allow us to *exactly* determine the correlators. Since we want to compare the results of this section with lattice data of the critical Ising spin chain with mixed BC, it will be particularly satisfying to find such equations for the cyclic orbifolds of the Ising BCFT.

Let's first consider the correlator:

$$\langle \sigma_{1,3}(z,\bar{z})\rangle_{N=2}^{\alpha\beta} \qquad (3.62)$$

in the $\mathbb{Z}_2$ Ising orbifold BCFT which should satisfy, up to a Möbius map, the same differential equation as:

$$\langle \Phi_{1,2}| \sigma_{1,3}(1)\sigma_{1,3}(\eta) |\Phi_{1,2}\rangle \qquad (3.63)$$

The orbifold fusion rules of [40], imply that the space of conformal blocks is two-dimensional since:

$$\sum_i \mathcal{N}^i_{\sigma_{1,3},\sigma_{1,3}} \mathcal{N}^{\Phi_{1,2}}_{i,\Phi_{1,2}} = 2 \tag{3.64}$$

By the same type of arguments and assumptions as in Section 3.5, we infer that (3.63) satisfies a second order Fuchsian differential equation with the following Riemann scheme:

| 0 | 1 | $\infty$ |
|---|---|---|
| $-h_{\sigma_{1,3}} - 2h_{1,2} + h_{\sigma_{\mathbf{1}}}$ | $-2h_{\sigma_{1,3}}$ | $h_{\sigma_{1,3}} - 2h_{1,2} + h_{\sigma_{1,1}}$ |
| $-h_{\sigma_{1,3}} - 2h_{1,2} + h_{\sigma_{1,3}} + \frac{1}{2}$ | $-2h_{\sigma_{1,3}} + 2h_{1,3}$ | $h_{\sigma_{1,3}} - 2h_{1,2} + h_{\sigma_{1,2}} + \frac{1}{2}$ |

so that we eventually find the one-point twist correlator to be

$$\langle \sigma_{1,3}(z,\bar{z}) \rangle^{\alpha\beta}_{(N=2)} = \bar{z}^{-2h_{\sigma_{1,3}}} g_+^{-1} \tilde{\mathcal{F}}^{N=2}_{\Psi_{\mathbf{1}}}(\eta) \tag{3.65}$$

with

$$\boxed{\tilde{\mathcal{F}}^{(N=2)}_{\Psi_{\mathbf{1}}}(\eta) = \frac{1 + \eta^{3/4}}{2(1-\eta)^{9/16}\eta^{3/8}}} \tag{3.66}$$

Finally, we can find an exact expression for the bare twist correlator:

$$\langle \sigma_{\mathbf{1}}(z,\bar{z}) \rangle^{\alpha\beta}_{N=3} \tag{3.67}$$

in the $\mathbb{Z}_3$ Ising orbifold BCFT since it also satisfies a second order differential equation with Riemann scheme:

| 0 | 1 | $\infty$ |
|---|---|---|
| $-h_{\sigma_{\mathbf{1}}} - 3h_{1,2} + h_{\sigma_{1,2}}$ | $-2h_{\sigma_{\mathbf{1}}}$ | $h_{\sigma_{\mathbf{1}}} - 3h_{1,2} + h_{\sigma_{1,2}}$ |
| $-h_{\sigma_{\mathbf{1}}} - 3h_{1,2} + h_{\sigma_{1,2}} + \frac{1}{3}$ | $-2h_{\sigma_{\mathbf{1}}} + 2h_{1,3}$ | $h_{\sigma_{\mathbf{1}}} - 3h_{\Psi_{1,2}} + h_{\sigma_{1,2}} + \frac{1}{3}$ |

We find:

$$\langle \sigma_{\mathbf{1}}(z,\bar{z}) \rangle^{\alpha\beta}_{N=3} = \bar{z}^{-1/9} g_+^{-2} \tilde{\mathcal{F}}^{N=3}_{\Psi_{\mathbf{1}}}(\eta) \tag{3.68}$$

with

$$\boxed{\tilde{\mathcal{F}}^{N=3}_{\Psi_{\mathbf{1}}}(\eta) = \frac{1 + \eta^{1/3}}{2(1-\eta)^{1/9}\eta^{1/6}}} \tag{3.69}$$

**Other results for the Ising BCFT.** We have also obtained results specific to the $\mathbb{Z}_2$ and $\mathbb{Z}_3$ orbifolds of the Ising BCFT with fixed mixed BC with $\alpha = +$ and $\beta = -$, for which the most relevant primary BCCO is $\psi^{(+-)}_{2,1}$. Since these results are not based on deriving differential equations, it felt thematically appropriate to leave their presentation for the Appendix D.

# 4 Numerical checks and finite-size corrections in quantum chains

To provide an independent appraisal of the validity of our CFT results, we have performed a numerical analysis on the Ising and three-state Potts open quantum chains for different settings of mixed BC. Once finite-size effects are properly accounted for, the validity of the CFT results becomes apparent.

We should note that the Rényi entropies in the Ising case have already been obtained, for generic $N$ in the work of [57], through a different approach. We found that our analytical calculations (for $N = 2, 3$) are compatible with their results.

Furthermore, by studying the finite-size corrections to their result, we manage to quantitatively understand the deviation of the chain data from the leading CFT prediction in the DMRG numerical analysis of [57], even for relatively large system sizes $M \sim 10^2$. Thus, when the subleading CFT contribution to the Rényi entropy is taken into account, as our analysis shall show, the agreement with the lattice data is excellent, even for the small system sizes $M \sim 26$ accessible to exact diagonalization.

## 4.1 The Ising quantum chain with mixed BC

The Hamiltonian of the Ising quantum chain with open BC, describing $M$ spins with generic BC at the boundary, is given by:

$$H_{\alpha\beta} = -\sum_{j=1}^{M-1} s_j^z s_{j+1}^z - h \sum_{j=1}^{M} s_j^x - h_\alpha s_1^z - h_\beta s_M^z \,, \tag{4.1}$$

where $s_j^{x,y,z}$ denote Pauli spin operators acting non-trivially at site $j$, and as identity at all the other sites. We denote the lattice spacing by $a$, so that the length of the chain is $L = Ma$. The parameters $h_\alpha, h_\beta$ denote external fields (in the $z$ direction) acting at the boundary sites $j = 1$ and $j = M$. The ground state of this Hamiltonian is then found by *exact diagonalization* (ED) for system sizes $M \leq 26$, and from it, the Rényi entropies are extracted.

To take the *scaling limit* of the critical chain, we send $M \to \infty, a \to 0$ while keeping $L$ fixed. In this limit, criticality is achieved in the bulk for $h = 1$, while each boundary admits three *critical points* $h_\alpha, h_\beta \in \{0, \pm\infty\}$.

From a CFT perspective, the scaling limit of the critical Ising chain with open boundaries is very well understood. It is described by the BCFT with central charge $c = 1/2$ and a bulk operator spectrum consisting of three primary operators – the identity $\mathbf{1}$, energy $\varepsilon$ and spin operators $s$ – and their descendants [77]. The three boundary critical points correspond to the three conformal boundary conditions for the Ising BCFT, which, in the framework of radial quantization on the annulus, allow the construction of the following physical boundary states [53, 77]:

$$|f\rangle = |\mathbf{1}\rangle\rangle - |\epsilon\rangle\rangle \qquad \text{(free BC)} \,, \tag{4.2}$$

$$|\pm\rangle = \frac{1}{\sqrt{2}}|\mathbf{1}\rangle\rangle + \frac{1}{\sqrt{2}}|\epsilon\rangle\rangle \pm \frac{1}{2^{1/4}}|s\rangle\rangle \qquad \text{(fixed BC)} \,, \tag{4.3}$$

where $|i\rangle\rangle$ denotes the Ishibashi state [53] [81] corresponding to the primary operator $i$. The physical boundary states $|\alpha\rangle$ are in one-to-one correspondence with the primary fields of the bulk CFT [2]: $|f\rangle \leftrightarrow s$ and $|\pm\rangle \leftrightarrow \mathbf{1}/\varepsilon$. The boundary fields that interpolate between two conformal BCs can be inferred from this correspondence, as shown in [53], [78]. Thus, the spectrum of primary boundary fields $\psi_i^{(\alpha\beta)}$ of the Ising BCFT is the one of Table 2.

On the discrete side, we are calculating the one-point correlator of the *lattice twist operator* $\hat{\sigma}(m, n)$, where $(m, n)$ are square-lattice coordinates. In the scaling limit with $a \to 0$, $\hat{\sigma}(m, n)$

---

[2]This statement is strictly true if the bulk CFT is diagonal, see [82] for a detailed discussion.

| $(\alpha\beta)$ | $+$ | $-$ | $f$ |
|---|---|---|---|
| $+$ | $\psi_{\mathbf{1}}$ | $\psi_{\varepsilon}$ | $\psi_s$ |
| $-$ | $\psi_{\varepsilon}$ | $\psi_{\mathbf{1}}$ | $\psi_s$ |
| $f$ | $\psi_s$ | $\psi_s$ | $\psi_{\mathbf{1}}, \psi_{\varepsilon}$ |

Table 2: Boundary operator spectrum of the Ising BCFT

admits a *local* expansion into scaling operators of the corresponding orbifold CFT. The two most *relevant* terms in this expansion are:

$$\widehat{\sigma}(m,n) = A\, a^{2h_\sigma} \sigma_{\mathbf{1}}(w,\bar{w}) + B\, a^{2h_{\sigma_\varepsilon}} \sigma_\varepsilon(w,\bar{w}) + \text{less relevant terms}\,, \qquad (4.4)$$

with the composite twist operator $\sigma_\varepsilon$ defined in (2.13) and $h_{\sigma_\varepsilon} = h_\sigma + h_\varepsilon/N$. The integers $(m,n)$ parametrize the lattice, and they are related to the continuum coordinate on the strip as $w = (m+in)a$, $\bar{w} = (m-in)a$. We can take advantage of the translation invariance in the $n$ direction to fix the "time" coordinate of the lattice twist operators to be $n = 0$. We will then denote their continuum coordinate by $\ell = ma$.

The amplitudes $A$ and $B$ in (4.4) are not universal quantities, so we cannot determine them by CFT techniques. However, they are also independent of the global properties of the system (e.g. choice of BC) so they can be found from a numerical analysis of the infinite Ising chain. Here one can employ the free fermion techniques of [1] and the well-known analytical results for the Rényi entropy of an interval in an infinite system [50,2] to fit for the values of $A$ and $B$, with great accuracy.

We can now express the lattice one-point twist correlator with generic mixed BC as an expansion of CFT correlators:

$$\langle \widehat{\sigma}(m,0) \rangle^{\alpha\beta} = A a^{2h_\sigma} \langle \sigma(\ell,\ell) \rangle^{\alpha\beta}_{\mathbb{S}_L} + B a^{2h_{\sigma_\varepsilon}} \langle \sigma_\varepsilon(\ell,\ell) \rangle^{\alpha\beta}_{\mathbb{S}_L} + \dots \qquad (4.5)$$

Using the map (3.2), we can make the dependence on system size in (4.5) explicit:

$$\langle \widehat{\sigma}(m,0) \rangle^{\alpha\beta} = A \left( \frac{M}{\pi} \right)^{-2h_\sigma} \langle \sigma(z,\bar{z}) \rangle^{\alpha\beta}_{\mathbb{H}} + B \left( \frac{M}{\pi} \right)^{-2h_{\sigma_\varepsilon}} \langle \sigma_\varepsilon(z,\bar{z}) \rangle^{\alpha\beta}_{\mathbb{H}} + \dots \qquad (4.6)$$

where $z = \exp(i\pi\ell/L), \bar{z} = \exp(-i\pi\ell/L)$. In our computational setup, the system sizes accessible through exact diagonalization are limited to $M \leq 26$ and, since twist operators are placed *between lattice sites*, we have only considered even system sizes.

With system sizes of this order of magnitude, finite-size corrections are quite strong. The most relevant corrections we have found arise from the subleading scaling of the lattice twist operator, given in equation (4.6). The relative scaling of the subleading term with respect to the leading one is $\mathcal{O}\left(M^{-2h_\epsilon/N}\right)$. Since we do not have access, numerically, to system sizes large enough to suppress these corrections, we had to take into account the first two terms in the expansion of (4.6) to find a good agreement with the lattice data. Furthermore, as the work of [57] suggests, the finite-size effects are still important, even at the much larger system sizes $M \sim 100$ accessible through DMRG methods. We mention that such subleading contributions to the lattice twist operator, which have been identified here from the operator spectrum of the $\mathbb{Z}_2$ cyclic orbifold, have previously been understood, through the path integral formalism on the corresponding replicated surface, under the name of "unusual corrections" [83,84].

We give now the results in the $\mathbb{Z}_2$ orbifold for the correlators appearing in the expansion (4.5), for *mixed fixed* BC with $\alpha = +, \beta = -$ (calculated in Appendix D) and mixed free-fixed

BC with $\alpha = +$ and $\beta = f$:

$$\langle \sigma_1(\ell, \ell) \rangle_{\mathbb{S}_L}^{+-} = 2^{-5/2} \left( \frac{2L}{\pi} \right)^{-1/16} \frac{7 + \cos \frac{2\pi\ell}{L}}{\left( \sin \frac{\pi\ell}{L} \right)^{1/16}},$$

$$\langle \sigma_\varepsilon(\ell, \ell) \rangle_{\mathbb{S}_L}^{+-} = 2^{-5/2} \left( \frac{2L}{\pi} \right)^{-9/16} \frac{1 - 9 \cos \frac{2\pi\ell}{L}}{\left( \sin \frac{\pi\ell}{L} \right)^{9/16}},$$

$$\langle \sigma_1(\ell, \ell) \rangle_{\mathbb{S}_L}^{+f} = 2^{1/2} \left( \frac{2L}{\pi} \right)^{-1/16} \frac{\cos \frac{\pi\ell}{4L}}{\left( \sin \frac{\pi\ell}{L} \right)^{1/16}},$$

$$\langle \sigma_\varepsilon(\ell, \ell) \rangle_{\mathbb{S}_L}^{+f} = -2^{1/2} \left( \frac{2L}{\pi} \right)^{-9/16} \frac{\cos \frac{3\pi\ell}{4L}}{\left( \sin \frac{\pi\ell}{L} \right)^{9/16}}, \tag{4.7}$$

where the interval $\ell$ starts at the $\alpha = +$ boundary. The expressions for the bare twist correlators are in accord with the equivalent results obtained in [57].

With this mention, we show in Figure 3 the remarkable agreement between our CFT calculations for the two terms contributing to the second Rényi entropy $S_2^{\alpha\beta} = -\log\langle \widehat{\sigma}(m, 0) \rangle^{(\alpha\beta)}$ of the interval $[0, m]$ on the lattice, and the numerical results for the critical Ising chain from the exact diagonalization of the Hamiltonian. Figure 3a illustrates the case of different ($\pm$) fixed BC on the two sides of the chain, while Figure 3b corresponds to letting the $m = 0$ site free, and applying a magnetic field at the boundary site $m = M - 1$.

To illustrate the large amplitude of finite-size effects, we show in Figure 4 how the CFT prediction fares against the lattice results with and without the incorporation of the subleading term. Even for the curve including both subleading and leading terms in (4.6), the agreement with lattice data is not perfect close to the boundary. This can be traced to the presence of corrections from *descendants* of twist operators, which introduce terms of $\mathcal{O}(M^{-h_\epsilon - 1})$ relative to the bare twist contribution.

We can repeat the same kind of analysis for the third Rényi entropy, related to the $\mathbb{Z}_3$-orbifold one-point function by $S_3^{\alpha\beta} = -\frac{1}{2} \log\langle \widehat{\sigma}(m, 0) \rangle^{(\alpha\beta)}$. The Ising orbifold correlators in this case are given by:

$$\langle \sigma_1(\ell, \ell) \rangle_{\mathbb{S}_L}^{+-} = 3^{-2} \left( \frac{2L}{\pi} \right)^{-1/9} \frac{7 + 2 \cos \frac{2\pi\ell}{L}}{\left( \sin \frac{\pi\ell}{L} \right)^{1/9}},$$

$$\langle \sigma_\varepsilon(\ell, \ell) \rangle_{\mathbb{S}_L}^{+-} = 3^{-2} \left( \frac{2L}{\pi} \right)^{-4/9} \frac{1 + 8 \cos \frac{2\pi\ell}{L}}{\left( \sin \frac{\pi\ell}{L} \right)^{4/9}},$$

$$\langle \sigma_1(\ell, \ell) \rangle_{\mathbb{S}_L}^{+f} = 2 \left( \frac{2L}{\pi} \right)^{-1/9} \frac{\cos \frac{\pi\ell}{3L}}{\left( \sin \frac{\pi\ell}{L} \right)^{1/9}},$$

$$\langle \sigma_1(\ell, \ell) \rangle_{\mathbb{S}_L}^{+f} = 2^{1/9} \left( \frac{2L}{\pi} \right)^{-4/9} \frac{\cos \frac{2\pi\ell}{3L}}{\left( \sin \frac{\pi\ell}{L} \right)^{4/9}}. \tag{4.8}$$

In Figure 5, we once again compare our CFT calculations (including both the leading and subleading term) with the critical chain results for the third Rényi entropy $S_3^{\alpha\beta}$, to good agreement for mixed fixed BC (Fig. 5a) and mixed free fixed BC (Fig. 5b). As for the $\mathbb{Z}_2$ results, including the CFT subleading contribution to $S_3^{\alpha\beta}$ is necessary to find a satisfying match with the lattice results. Further finite-size corrections in this case decay as $\mathcal{O}\left( M^{-\frac{2h_\varepsilon}{3} - 1} \right)$.

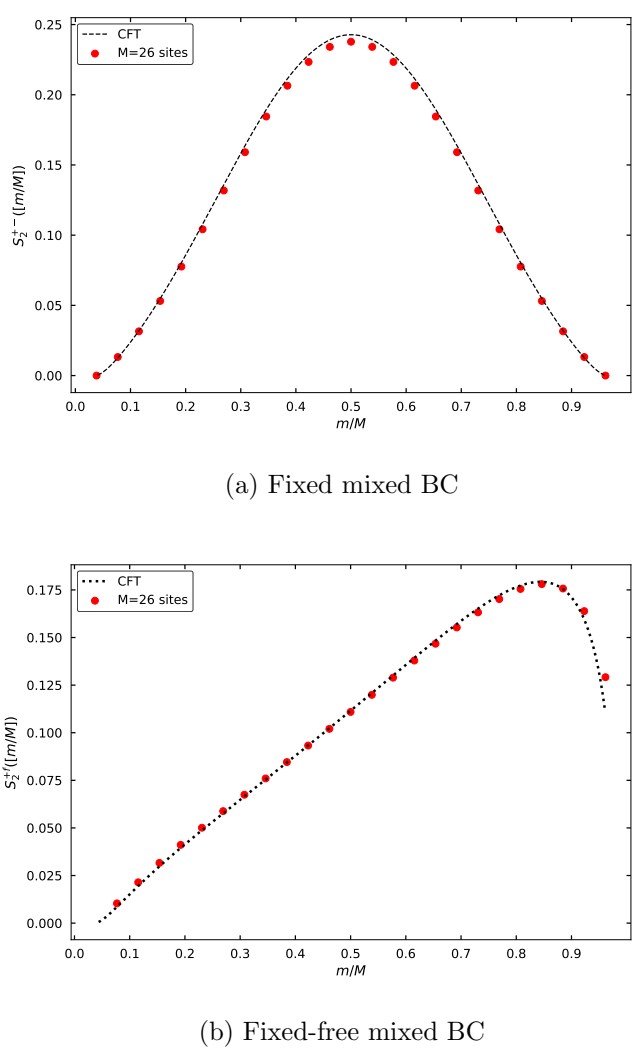

(a) Fixed mixed BC

(b) Fixed-free mixed BC

Figure 3: Plots of the second Rényi entropy $S_2^{\alpha\beta}([m/M])$ in the critical Ising chain with two types of mixed BC for a chain of size $M = 26$. The interval is grown from the $\beta = +$ boundary.

As advertised in the beginning of the section, our results for the bare twist correlators (for all configurations of mixed BC) are compatible with the ones of [57]. The subleading contribution to the Rényi entropies from the excited twist correlator is largely responsible for the mismatch between the lattice and CFT data in the aforementioned article. Finite-size corrections of this magnitude can be suppressed only with much larger system sizes $M \sim 10^3$, as the authors of the present work have shown in [60] .

## 4.2 The three-state Potts quantum chain with mixed BC

A natural extension of the Ising chain, the three-state Potts model allows the spins at each site to take one of three possible values $\{R, G, B\}$, which we can also conveniently parametrize by third roots of unity $\{1, \omega, \omega^2\}$, with $\omega = \exp(2\pi i/3)$. The Hamiltonian of the three-state Potts

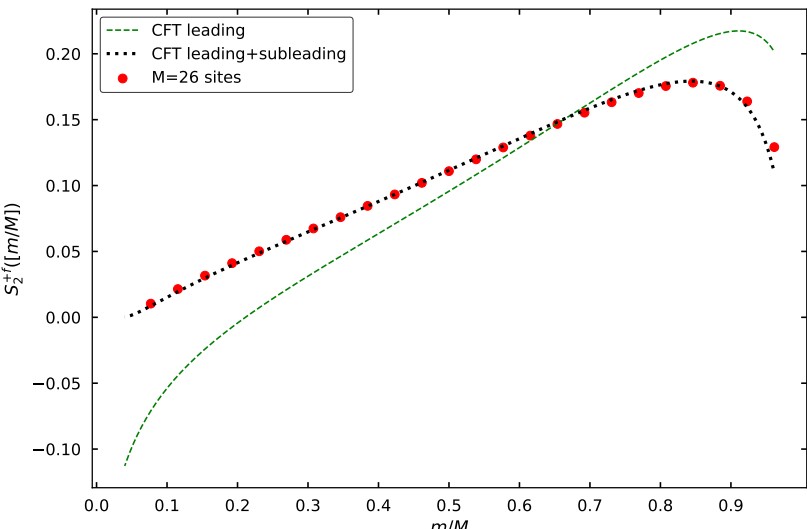

Figure 4: Comparison of the second Rényi entropy in the critical Ising chain of size $M = 26$ with mixed free fixed BC with CFT results. Inclusion of the subleading term in the expansion 4.6 is crucial for obtaining a satisfying agreement with lattice data

model, tuned to its bulk critical point [85] [69], [86] is given by:

$$H_{\alpha\beta} = -\zeta \left[ \sum_{j=1}^{M-1} \left( Z_j Z_{j+1}^\dagger + Z_j^\dagger Z_{j+1} \right) + \sum_{j=2}^{M-1} \left( X_j + X_j^\dagger \right) + H_1^{(\alpha)} + H_M^{(\beta)} \right], \qquad (4.9)$$

where $\zeta = \frac{\sqrt{3}}{2\pi^{3/2}}$ is the conformal normalization factor [86] and the operators $Z_j$ and $X_j$ act at site $j$ as:

$$Z = \begin{pmatrix} 1 & 0 & 0 \\ 0 & \omega & 0 \\ 0 & 0 & \omega^2 \end{pmatrix}, \qquad X = \begin{pmatrix} 0 & 1 & 0 \\ 0 & 0 & 1 \\ 1 & 0 & 0 \end{pmatrix}. \qquad (4.10)$$

The terms $H_1^{(\alpha)}$ and $H_M^{(\beta)}$ set the BCs at the ends of the chain. For the purpose of this analysis, we will set *fixed* BC of type $R$ at site 1 and *restricted* boundary conditions of type $\{G, B\}$ at site $M$ – the spin at site $M$ is forbidden from taking the value $R$. This is implemented through the boundary terms:

$$H^{(R)} = h \begin{pmatrix} -1 & 0 & 0 \\ 0 & 0 & 0 \\ 0 & 0 & 0 \end{pmatrix}, \qquad H^{(GB)} = h \begin{pmatrix} 1 & 0 & 0 \\ 0 & 0 & -1 \\ 0 & -1 & 0 \end{pmatrix}, \qquad (4.11)$$

The critical points of interest for the boundaries correspond to $h = +\infty$. However, for any $h > 0$, the boundaries will flow towards the same critical points, up to irrelevant boundary perturbations [87] . These are typically inconsequential for $h$ a large positive value. Furthermore, in our numerical analysis we can, in fact, implement $|h| \to \infty$ by restricting the local Hilbert spaces of the boundary sites to exclude the $\{G, B\}$ and $\{R\}$ configurations on the left and, respectively, right boundary.

The scaling limit $M \to \infty$, $a \to 0$ (with $L = Ma$ fixed) of this critical chain is also well understood. It is given by the D-series BCFT $\mathcal{M}_{6,5}$ with central charge $c = 4/5$ and a bulk

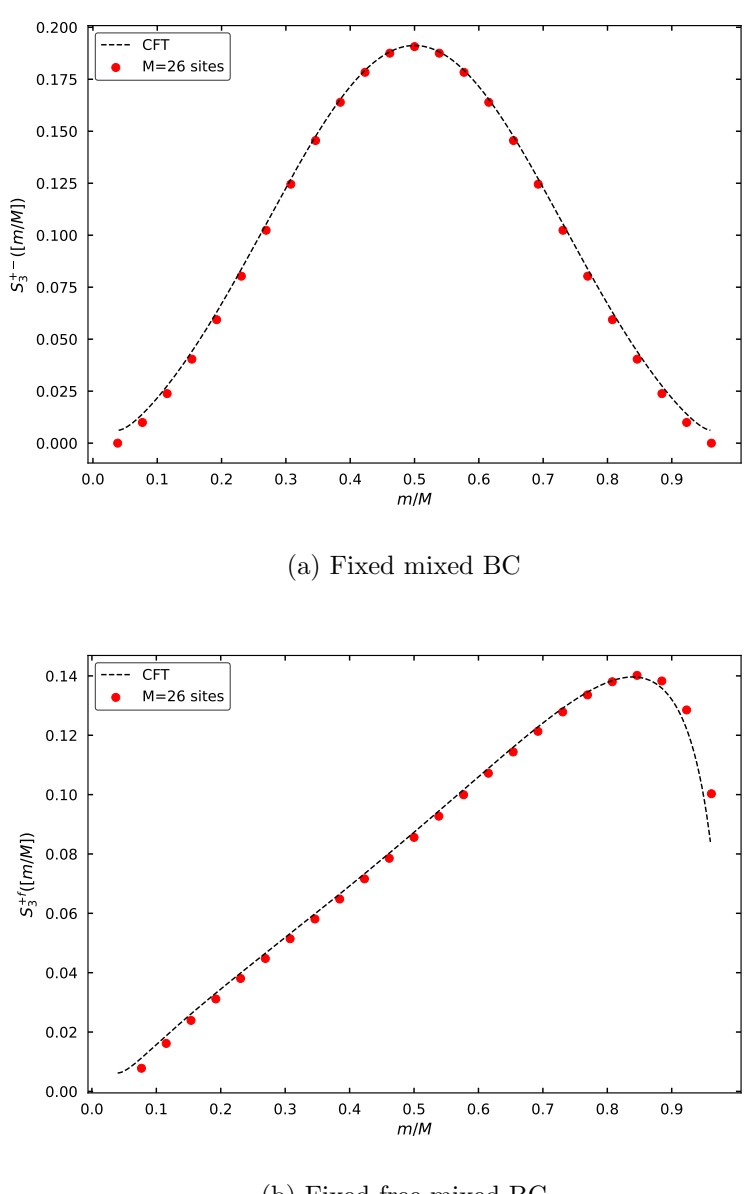

(a) Fixed mixed BC

(b) Fixed-free mixed BC

Figure 5: Plots of the third Rényi entropy $S_3^{\alpha\beta}([m/M])$ in the critical Ising chain with two types of mixed BC for a chain of size $M = 26$. The interval is grown from the $\alpha = +$ boundary.

primary operator spectrum that contains the scalar fields given in Table 3 as well as the non-diagonal fields $\{\phi_{2/5,7/5}, \phi_{7/5,2/5}, \phi_{3,0}, \phi_{0,3}\}$ whose labels indicate their respective holomorphic and antiholomorphic conformal dimensions. One can, as shown in Table 1, assign a $\mathbb{Z}_3$ charge to the scalar fields, and their respective conformal families, that is consistent with the fusion rules between them. The $\dagger$ in Table 3 is, thusly, used to differentiate the fields with the same conformal dimension, but different $\mathbb{Z}_3$ charge.

In the scaling limit, the fixed and restricted boundary critical points will correspond, natu-

| Diagonal fields | $(h, \bar{h})$ | $\mathbb{Z}_3$ charge |
|---|---|---|
| $\mathbf{1}$ | $(0, 0)$ | $0$ |
| $\varepsilon \equiv \phi_{1,2}$ | $(\frac{2}{5}, \frac{2}{5})$ | $0$ |
| $\phi_{1,3}$ | $(\frac{7}{5}, \frac{7}{5})$ | $0$ |
| $\phi_{1,4}$ | $(3, 3)$ | $0$ |
| $s, s^\dagger \equiv \phi_{3,3}$ | $(\frac{1}{15}, \frac{1}{15})$ | $\pm 1$ |
| $\psi, \psi^\dagger \equiv \phi_{3,4}$ | $(\frac{2}{3}, \frac{2}{3})$ | $\pm 1$ |

Table 3: Spectrum of spinless primary operators in the three-state Potts CFT

rally, to the fixed and restricted[3] conformal boundary states [53, 88].

$$
\begin{aligned}
|\mathbf{1}\rangle &= \mathcal{N}[(|\mathbf{1}\rangle\!\rangle + |\psi\rangle\!\rangle + |\psi^\dagger\rangle\!\rangle) + \lambda(|\epsilon\rangle\!\rangle + |s\rangle\!\rangle + |s^\dagger\rangle\!\rangle)] && \text{(fixed } R) \\
|\psi\rangle &= \mathcal{N}[(|\mathbf{1}\rangle\!\rangle + \omega|\psi\rangle\!\rangle + \bar{\omega}|\psi^\dagger\rangle\!\rangle) + \lambda(|\epsilon\rangle\!\rangle + \omega|s\rangle\!\rangle + \bar{\omega}|s^\dagger\rangle\!\rangle)] && \text{(fixed } G) \\
|\psi^\dagger\rangle &= \mathcal{N}[(|\mathbf{1}\rangle\!\rangle + \bar{\omega}|\psi\rangle\!\rangle + \omega|\psi^\dagger\rangle\!\rangle) + \lambda(|\epsilon\rangle\!\rangle + \bar{\omega}|s\rangle\!\rangle + \omega|s^\dagger\rangle\!\rangle)] && \text{(fixed } B) \\
|\varepsilon\rangle &= \mathcal{N}[\lambda^2(|\mathbf{1}\rangle\!\rangle + |\psi\rangle\!\rangle + |\psi^\dagger\rangle\!\rangle) - \lambda^{-1}(|\epsilon\rangle\!\rangle + |s\rangle\!\rangle + |s^\dagger\rangle\!\rangle)] && \text{(restricted } GB) \\
|s\rangle &= \mathcal{N}[\lambda^2(|\mathbf{1}\rangle\!\rangle + \omega|\psi\rangle\!\rangle + \bar{\omega}|\psi^\dagger\rangle\!\rangle) - \lambda^{-1}(|\epsilon\rangle\!\rangle + \omega|s\rangle\!\rangle + \bar{\omega}|s^\dagger\rangle\!\rangle)] && \text{(restricted } RB) \\
|s^\dagger\rangle &= \mathcal{N}[\lambda^2(|\mathbf{1}\rangle\!\rangle + \bar{\omega}|\psi\rangle\!\rangle + \omega|\psi^\dagger\rangle\!\rangle) - \lambda^{-1}(|\epsilon\rangle\!\rangle + \bar{\omega}|s\rangle\!\rangle + \omega|s^\dagger\rangle\!\rangle)] && \text{(restricted } RG) ,
\end{aligned}
\tag{4.12}
$$

where

$$
\mathcal{N} = \sqrt{\frac{2}{\sqrt{15}} \sin\frac{\pi}{5}}, \qquad \lambda = \sqrt{\frac{\sin(2\pi/5)}{\sin(\pi/5)}}, \tag{4.13}
$$

and the $|i\rangle\!\rangle$'s are the Ishibashi states defined in [53]. These conformal boundary states are labelled by the primary fields of Table 3.

Due to the $\mathbb{Z}_3$ symmetry of our model, we have some freedom to set which conformal boundary state corresponds to the *fixed* boundary condition $R$ in the chain. However, this uniquely determines the CFT boundary state that corresponds to the *restricted* boundary conditions $GB$. This can be understood by considering the spectrum of boundary fields that can interpolate between these conformal BC [56], and ensuring the results are consistent with the underlying $\mathbb{Z}_3$ symmetry . In our case, choosing *fixed* $R \leftrightarrow |\mathbf{1}\rangle$ forces us to assign *restricted* $GB \leftrightarrow |\varepsilon\rangle$. The most relevant boundary field interpolating between these BCs is $\psi_{1,2}^{(R,GB)}$ [56] with conformal dimension $h_\varepsilon = 2/5$.

We will now compare the quantum chain data for the second Rényi entropy in the critical Potts chain with our correlator calculations in the $\mathbb{Z}_2$ orbifold of the BCFT defined above. Our analysis will parallel the one for the Ising critical chain. We first hypothesize the form of the local expansion (4.4) of the lattice twist operator $\hat{\sigma}_{m,n}$ in the case of the three-state Potts model:

$$
\hat{\sigma}(m, n) = A\, a^{2h_\sigma} \sigma(w, \bar{w}) + B\, a^{2h_{\sigma_\varepsilon}} \sigma_\varepsilon(w, \bar{w}) + \text{less relevant terms}, \tag{4.14}
$$

where $h_\sigma = 1/20$, and the composite twist operator $\sigma_\varepsilon$ is built with the energy operator $\varepsilon$ of the Potts model so that $h_{\sigma_\varepsilon} = 1/4$. Once again, we numerically estimated the parameters $A, B$ by a simple analysis on the critical three-state Potts critical chain with *periodic boundary conditions*. Following this, the one-point lattice twist correlator with our choice of mixed BC can be calculated from:

$$
\langle \hat{\sigma}(m, 0) \rangle^{(GB,R)} = A \left(\frac{M}{\pi}\right)^{-2h_{\sigma_\mathbf{1}}} \langle \sigma(z, \bar{z}) \rangle_\mathbb{H}^{(GB,R)} + B \left(\frac{M}{\pi}\right)^{-2h_{\sigma_\varepsilon}} \langle \sigma_\varepsilon(z, \bar{z}) \rangle_\mathbb{H}^{(GB,R)} + \dots \tag{4.15}
$$

---

[3]In [53] they are referred to as "mixed" BC.

The correlators in (4.15) satisfy the second order (3.10) and fourth order (3.30) ODEs with $g = 6/5$. While the solutions to equation (3.10) are known exactly (3.13), one needs to solve (3.30) numerically to find the conformal blocks in the expansion (3.4) of the excited twist correlator $\langle \sigma_\varepsilon(z, \bar{z}) \rangle^{\alpha\beta}_{\mathbb{H}}$. This is done by a standard numerical implementation of the Frobenius method, whose details we leave for Appendix H.

As in the case of the Ising BCFT, not all the solutions of these differential equations are needed to build the twist field correlators in (4.15). Crucially, we note that in the three-state Potts mother BCFT, there is no boundary operator $\psi^{(RR)}_{7/5}$ living on the fixed conformal boundary of type $R$ [56]. At the level of the operator algebra this translates into the vanishing of the boundary-boundary structure constants $B^{(R|R|GB),\psi_{1,2}}_{\psi_{7/5},\psi_{1,2}}$ as we have checked using the results of [82]. This implies, through the relations between mother BCFT and orbifold structure constants derived in Section 2.3, the vanishing of some of the coefficients in the block expansions (3.4) of the correlators in (4.15). In effect, only the block corresponding to the identity operator contributes to these expressions when the twist fields are sent to the $\beta$ boundary. It corresponds to the following fusion rules for $\sigma_\mathbf{1}$, $\sigma_\varepsilon$:

$$\sigma_\mathbf{1}\bigg|_\beta \to \Psi^{(\beta\beta)}_\mathbf{1} \qquad \sigma_\varepsilon\bigg|_\beta \to \Psi^{(\beta\beta)}_\mathbf{1} \tag{4.16}$$

Thusly, we are led to obtain expressions for the bare twist and excited twist correlators on the UHP:

$$\langle \sigma(z, \bar{z}) \rangle^{\alpha\beta} = \bar{z}^{-2h_\sigma} \mathcal{A}^{(\beta)}_{\sigma,\Psi_\mathbf{1}} \mathcal{B}^{(\beta\beta\alpha)\Psi_{12}}_{\Psi_\mathbf{1},\Psi_{12}} \tilde{\mathcal{F}}_\mathbf{1}(\eta),$$

$$\langle \sigma_\varepsilon(z, \bar{z}) \rangle^{\alpha\beta} = \bar{z}^{-2h_{\sigma_\varepsilon}} \mathcal{A}^{(\beta)}_{\sigma_\varepsilon,\Psi_\mathbf{1}} \mathcal{B}^{(\beta\beta\alpha)\Psi_{12}}_{\Psi_\mathbf{1},\Psi_{12}} \tilde{\mathcal{F}}^{(\varepsilon)}_\mathbf{1}(\eta), \tag{4.17}$$

where $\tilde{\mathcal{F}}_\mathbf{1}(\eta)$ is given in (3.13), so in this case we can write an explicit result for the bare twist correlator on the UHP:

$$\boxed{\langle \sigma(z, \bar{z}) \rangle^{\alpha\beta} = \mathcal{A}^{(\beta)}_{\sigma,\Psi_\mathbf{1}} \mathcal{B}^{(\beta\beta\alpha)\Psi_{12}}_{\Psi_\mathbf{1},\Psi_{12}} (1-\eta)^{-2h_\sigma} \eta^{-2h_{12}+h_\sigma} {}_2\mathrm{F}_1\left(-8/5, -9/10; -9/5 \mid 1-\eta\right)} \tag{4.18}$$

For the excited twist correlator we have:

$$\tilde{\mathcal{F}}^{(\varepsilon)}_\mathbf{1}(\eta) = J_1(u(\eta)) = (1-u)^{-2h_{\sigma_\varepsilon}} \sum_{n=0}^\infty a_n(1-u)^n, \tag{4.19}$$

with the coefficients determined by the recursion relation (H.8), derived in Appendix H.

The structure constants can be expressed in terms of known quantities for the $\mathcal{M}(6,5)$ BCFT, also obtained in Appendix B:

$$\mathcal{A}^{(\beta)}_{\sigma_\varepsilon,\Psi_\mathbf{1}} = g_R^{-1} A_\varepsilon^R, \qquad \mathcal{A}^{(\beta)}_{\sigma,\Psi_\mathbf{1}} = g_R^{-1}, \qquad \mathcal{B}^{(\beta\beta\alpha)\Psi_{12}}_{\Psi_\mathbf{1},\Psi_{12}} = 1, \tag{4.20}$$

where the ground state degeneracies $g_R$ and $g_{GB}$ have been found in [69]:

$$g_R = \left(\frac{5-\sqrt{5}}{30}\right)^{\frac{1}{4}} \qquad g_{GB} = g_R \lambda^2 \tag{4.21}$$

and the bulk-boundary structure constant $A_\varepsilon^R$ has been calculated in [88, 89] to be:

$$A_\varepsilon^R = \left(\frac{1+\sqrt{5}}{2}\right)^{\frac{3}{2}}. \tag{4.22}$$

Putting everything together, we can finally compare the lattice prediction for the second Rényi entropy $S_2^{(R,GB)} = -\log\langle\hat\sigma(m,0)\rangle^{(R,GB)}$ with our analytic results in Figure 6. While the CFT prediction does not satisfyingly match the lattice data at all points, we observe that the inclusion of the subleading term gives an analytic curve that is closer to the lattice data. However, it is not enough to make up for the severe finite-size effects.

Firstly, due to the operator content of the D-series $\mathcal{M}_{6,5}$ CFT, we expect the higher order corrections in 4.15 to have a slower power law decay than in the case of the Ising CFT. We conjecture that the next-to-subleading contribution to (4.15) will decay as $\sim M^{-2(h_{\sigma_\varepsilon}+1/2)}$. These corrections, we believe, arise from the combined contribution of the $\langle\sigma_{\phi_{1,3}}(w,\bar w)\rangle_{\mathbb{S}}^{R,GB}$ and $\langle L_{-1/2}^{(1)}\bar L_{-1/2}^{(1)}\sigma_{\phi_{1,2}}(w,\bar w)\rangle_{\mathbb{S}}^{R,GB}$. While the first correlator can be calculated by a repeat of the method employed for the subleading term, the correlator involving the descendant twist field requires the derivation of a new differential equation. Such an endeavour is beyond the scope of this work.

Furthermore, the quantum chain sizes we can reach are diminished in the case of the three-state Potts model, since the size of the space of states grows as $\sim 3^M$. This memory constraint prevents us from reaching sizes at which higher order corrections are suppressed, using our computational methods. This limitation can be, perhaps, bypassed through the usage of more sophisticated numerical tools, such as DMRG or tensor network methods, to access system sizes $M$ for which the unknown higher-order correction terms are further suppressed.

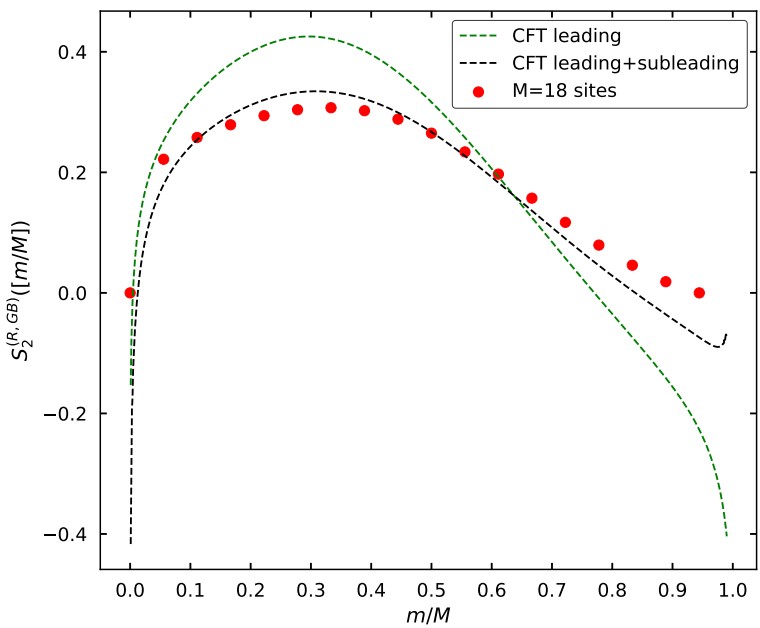

Figure 6: Comparison of the second Rényi entropy in the critical three-state Potts chain of size $M = 18$ with mixed $(R, GB)$ BC with CFT results.

Finally, one can use the method of Appendix H, applied this time to the third order ODE of Section 3.4 to derive the leading CFT contribution to the $S_3^{(R,GB)}([0,\ell])$ Rényi entropy. Since in this case, we have not derived an ODE for the excited twist correlator, we have no handle on the finite-size corrections to the lattice data, which should be even more severe for $N = 3$.

Instead, we have just checked that the CFT result for mixed BC interpolates between the third Rényi entropies for identical $R$ and $GB$ boundaries:

$$S_3^{(\alpha,\alpha)}([0,\ell]) = \frac{c}{9} \log \left[ \frac{2L}{\pi} \sin \left( \frac{\pi\ell}{L} \right) \right] + \log g_\alpha \tag{4.23}$$

Our expectations are met, as Figure 7 confirms.

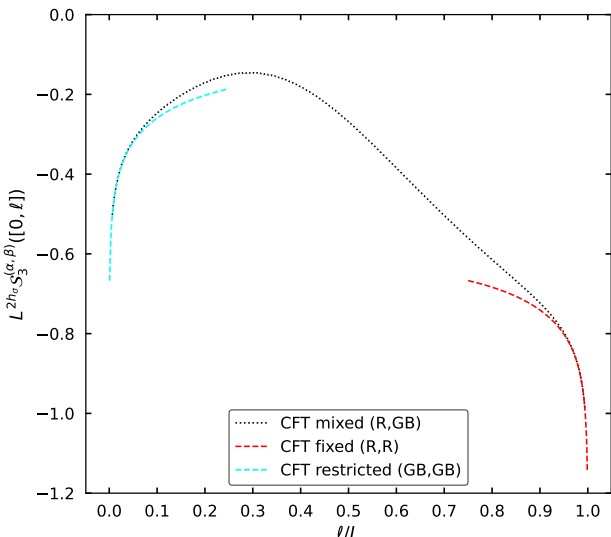

Figure 7: Comparison of shifted third Rényi entropies for $(R, GB)$, $(R, R)$ and $(GB, GB)$ BC. The mixed BC curve can be seen to interpolate between the identical BC results

# 5    Conclusion

In this article, we have presented a general method for calculating Rényi entropies $S_N$ in the ground state of a 1D critical system with mixed open boundaries, for an interval starting at one of its ends. This required computing three-point functions of one twist operator and two BCCOs on the upper-half plane $\mathbb{H}$ with mixed BCs $(\alpha, \beta)$ in the $\mathbb{Z}_N$ cyclic orbifold.

For this purpose, we have derived ODEs satisfied by these correlation functions, by exploiting the null-vectors of the twisted and untwisted representations of its symmetry algebra $\mathrm{OVir}_N$, together with Ward identities obtained from the additional conserved currents of the theory. We used a combination of analytical and numerical methods to find a basis of solutions (*a.k.a* conformal blocks) of these ODEs.

For the examples provided in this work, we have calculated the boundary and bulk-boundary structure constants needed to build the physical correlators as linear combinations of the blocks. Among the setups we have analysed are the leading and subleading contributions to the one-interval second and third Rényi entropies of the Ising model, and the second Rényi entropy for the three-state Potts model. We have also derived differential equations for mixed BC twist field correlators in the $\mathbb{Z}_2$ and $\mathbb{Z}_3$ orbifolds of generic BCFTs, and obtained an explicit expression for the second Rényi entropy valid for any diagonal minimal model, but with a particular set of mixed boundary conditions.

We have compared the CFT results against critical Ising and three-state Potts spin chain data. Since finite size effects are quite significant for open chains, we have included both the leading and subleading contributions to the lattice twist field correlator in our analytical prediction. In the Ising case, the agreement was excellent for all choices of mixed BC, even though the system sizes we could reach were limited. For the three-state Potts chain, however, the finite size effects are even more severe, and as a consequence the matching is less satisfactory. This could be improved by using more sophisticated numerical techniques such as DMRG [90,91] or tensor network methods [92].

The clearest limitation of our method, first identified in [42], is that the process for obtaining a differential equation becomes more difficult as $N$ is increased. We have checked using the fusion rules in Appendix F for $N > 3$ that the expected order of the ODEs increases with $N$ for generic minimal models, which implies that more orbifold Ward identities will be needed to obtain the ODEs.

There are several possible extensions of this work. A possibility would be to generalize the setup for the calculation of Rényi entropies of an interval *contained in the bulk*, with mixed BC. However, in this situation, one would have to find a differential equation that a four-point function with two twist fields and two BCCOs satisfies. Cardy's doubling trick suggests that such a correlator satisfies the same Ward identities as a six-point conformal block on the complex plane, so the corresponding differential equation would be partial instead of ordinary.

# Appendix

## A    Mother BCFT conventions

We will define here our mother BCFT conventions on the upper-half plane $\mathbb{H}$ parametrized by the coordinate $z = x + iy$. The boundary is aligned with the real axis.

Bulk operators in the mother CFT are denoted by $\phi_i(z, \bar{z})$ while boundary operators are written as $\psi_j^{(ab)}(x)$. The operator algebra consists of three types of OPE, which we explicitate, to fix the notations for the corresponding structure constants.

First, we have the *bulk-bulk OPEs*:

$$\phi_i(z, \bar{z})\phi_j(0,0) = \sum_{\phi_k \text{ scaling op.}} C_{ij}^k z^{-h_i - h_j + h_k} \bar{z}^{-\bar{h}_i - \bar{h}_j + \bar{h}_k} \, \phi_k(0,0) \tag{A.1}$$

where $C_{ij}^k$ are the *bulk structure constants*.

The second type of OPE are **boundary-boundary OPEs** between BCCOs interpolating different boundary conditions:

$$\psi_i^{(ab)}(x)\psi_j^{(dc)}(y) = \delta_{bd} \sum_k B_{\psi_i \psi_j}^{(abc)\psi_k}(x - y)^{h_k - h_i - h_j} \psi_k^{(ac)}(y) \tag{A.2}$$

for $x > y$. The $B_{\psi_i \psi_j}^{(abc)\psi_k}$ are the *boundary-boundary structure constants*. The Kronecker delta formally expresses the fact that it only makes sense to consider correlations of boundary operators ordered such that their BCs change consistently with their labelling.

Finally, we consider the third kind of OPE, between the bulk and the boundary:

$$\phi_i(z) = \sum_k A_{\phi_i, \psi_k}^{(a)}(2y)^{h_k - \Delta_i} \cdot \psi_k^{(aa)}(x) \tag{A.3}$$

with $A_{\phi_i, \psi_k}^{(a)}$ the bulk-boundary structure constants.

In [93], [82] all the structure constants $A_{\phi_i, \psi_k}^{(a)}$ and $B_{\psi_i \psi_j}^{(abc)\psi_k}$ have been determined for A-series and D-series BCFTs, in terms of fusion matrix elements of bulk CFT four-point functions, and the entries of the modular $S$ matrix. Relevant for this paper are the results:

$$\boxed{B_{\psi_i \psi_j}^{(abc)\psi_k} = \mathbf{F}_{bk} \begin{bmatrix} a & c \\ i & j \end{bmatrix}} \tag{A.4}$$

where the fusion matrix relates bases of conformal blocks around $z = 0$ and $z = 1$

$$\mathcal{I}_{ia,cj}^r(z) = \sum_{rs} \mathbf{F}_{rs} \begin{bmatrix} a & c \\ i & j \end{bmatrix} \mathcal{J}_{ij,ac}^s(1 - z) \tag{A.5}$$

defined in the bulk.

We also give the expressions for the 1-point structure constants of the BCFT in terms of $S$-matrix elements of the mother CFT

$$\boxed{A_{\phi_i}^{(a)} \equiv A_{\phi_i, \psi_1}^{(a)} = \frac{S_{ai}}{S_{a1}}\sqrt{\frac{S_{11}}{S_{i1}}}} \tag{A.6}$$

# B Computation of orbifold structure constants

## B.1 Composite twist one-point structure constant in the $\mathbb{Z}_N$ orbifold BCFT

Assuming the one-point structure constant $\mathcal{A}^{(\alpha)}_{\sigma_1,\psi_1}$ is known, let's consider the correlator:

$$\langle \sigma_j(0,0) \rangle^\alpha_\mathbb{D} = \mathcal{A}^{(\alpha)}_{\sigma_j,\psi_1} \tag{B.1}$$

We now use (2.13) to write the LHS of (B.1) as:

$$\langle \sigma_j(0,0) \rangle^\alpha_\mathbb{D} = \mathcal{A}_j \lim_{\epsilon \to 0} \epsilon^{2(1-N^{-1})h_j} \left\langle \Phi_{[j,1,\dots,1]}(\epsilon,\bar\epsilon)\sigma^{[k]}(0,0) \right\rangle^\alpha_\mathbb{D} \tag{B.2}$$

Substituting the definition (2.9) of non-diagonal fields, we find:

$$\langle \sigma_j(0,0) \rangle^\alpha_\mathbb{D} = N^{-2(1-N^{-1})h_j-1} \lim_{\epsilon \to 0} \epsilon^{2(1-N^{-1})h_j} \sum_{a=0}^{N-1} \left\langle \left(\phi_{j+a} \otimes \phi_{1+a} \otimes \dots \phi_{1+a}\right)(\epsilon,\bar\epsilon)\sigma^{[k]}(0,0) \right\rangle^\alpha_\mathbb{D} \tag{B.3}$$

Each correlator in the sum above can be written as:

$$\left\langle \left(\phi_{j+a} \otimes \phi_{1+a} \otimes \dots \phi_{1+a}\right)(\epsilon,\bar\epsilon)\sigma^{[k]}(0,0) \right\rangle^\alpha_\mathbb{D} = \frac{Z_{N,a}}{Z^N_{1,a}} \langle \phi_j(\epsilon,\bar\epsilon) \rangle_{\mathbb{D}_N} = \mathcal{A}^{(\alpha)}_{\sigma_1,\psi_1} \langle \phi_j(\epsilon,\bar\epsilon) \rangle^a_{\mathbb{D}_N} \tag{B.4}$$

where $Z_{N,a}$ denotes the partition function on the $N$-sheeted disk with conformal BC $a$, and branch point at 0. Now, we can unfold the disk correlator through the conformal map $w \to w^{1/N}$ and substitute back in (B.3) to find:

$$\langle \sigma_j(0,0) \rangle^\alpha_\mathbb{D} = \mathcal{A}^{(\alpha)}_{\sigma_1,\psi_1} \langle \phi_j(0,0) \rangle^a_\mathbb{D} \tag{B.5}$$

so that we finally find:

$$\boxed{\mathcal{A}^{(\alpha)}_{\sigma_j,\psi_1} = \mathcal{A}^{(\alpha)}_{\sigma_1,\psi_1} A^a_{\phi_j}} \tag{B.6}$$

## B.2 Bulk-boundary structure constant in the $\mathbb{Z}_2$ orbifold CFT

In this section we compute the structure constant $\mathcal{A}^{(\alpha)}_{\sigma_1,\Psi_{1,3}}$, which is given by the UHP correlator:

$$\left\langle \sigma_1(i/2,-i/2)\Psi^{(\alpha\alpha)}_{1,3}(1) \right\rangle^\alpha_\mathbb{H} = \mathcal{A}^{(\alpha)}_{\sigma_1,\Psi_{1,3}} \tag{B.7}$$

We can map the LHS of (B.7) to the unit disk through:

$$z \to \frac{z-i/2}{z+i/2} \tag{B.8}$$

and then use the partition function expression of the correlator (as in the previous section) to find (after a global rotation):

$$\left\langle \sigma_1(0,0)\Psi^\alpha_{1,3}(-i) \right\rangle_\mathbb{D} = \langle \sigma_1(0,0) \rangle^\alpha_\mathbb{D} \left\langle \psi^\alpha_{1,3}(-i)\psi^\alpha_{1,3}(-ie^{2i\pi}) \right\rangle_{\mathbb{D}_{2,a}} \tag{B.9}$$

where $D_{2,a}$ is a 2-sheeted disk with branch point at 0. We unfold the correlator of boundary fields through the map $w \to w^{1/2}$ to find:

$$\left\langle \psi^\alpha_{1,3}(-i)\psi^\alpha_{1,3}(-ie^{2i\pi}) \right\rangle_{\mathbb{D}_{2,a}} = \left(2i^{-1/2}\right)^{-2h_{1,3}} \left\langle \psi^{(aa)}_{1,3}(i^{1/2})\psi^{(aa)}_{1,3}(-i^{1/2}) \right\rangle_\mathbb{D} = 1 \tag{B.10}$$

so that, by putting everything together, we arrive at:

$$\boxed{\mathcal{A}^{(\alpha)}_{\sigma_1,\Psi_{1,3}} = \mathcal{A}^{(\alpha)}_{\sigma_1,\Psi_1}}$$ (B.11)

For generic $N$, expressing the bulk-boundary structure constant $\mathcal{A}^{(\alpha)}_{\sigma_1,\Psi_{1,3}}$ in terms of mother BCFT quantities depends on our ability to calculate $N$-point functions of boundary operators. For $N \geq 5$, this becomes difficult to solve for generic mother BCFTs.

## C  Orbifold Ward identities for bulk fields

Following [42], we give here the **orbifold Ward identities for 4-point bulk correlators**:

$$\sum_{p=0}^{\infty} a_p \left\langle \mathcal{O}_1 \left| L^{(r)}_{-m_1-p} \mathcal{O}_2(1)\mathcal{O}_3(x,\bar{x}) \right| \mathcal{O}_4 \right\rangle = \sum_{p=0}^{\infty} b_p \left\langle \mathcal{O}_1 \left| \left[ L^{(r)}_{m_2+p}\mathcal{O}_2 \right](1)\mathcal{O}_3(x,\bar{x}) \right| \mathcal{O}_4 \right\rangle$$

$$+ \sum_{p=0}^{\infty} c_p \left\langle \mathcal{O}_1 \left| \mathcal{O}_2(1) \left[ L^{(r)}_{m_3+p}\mathcal{O}_3 \right](x,\bar{x}) \right| \mathcal{O}_4 \right\rangle \quad \text{(C.1)}$$

$$+ \sum_{p=0}^{\infty} d_p \left\langle \mathcal{O}_1 \left| \mathcal{O}_2(1)\mathcal{O}_3(x,\bar{x}) L^{(r)}_{m_4+p} \right| \mathcal{O}_4 \right\rangle$$

where the *levels $m_i \in \mathbb{Z} + rk_i/N$* satisfy:

$$m_1 + m_2 + m_3 + m_4 = -2$$ (C.2)

and the coefficients $a_p$, $b_p$, $c_p$ and $d_p$ are defined from the Taylor series:

$$(1-z)^{m_2+1}(1-xz)^{m_3+1} = \sum_{p=0}^{\infty} a_p z^p$$ (C.3)

$$(z-x)^{m_3+1} z^{m_4+1} = \sum_{p=0}^{\infty} b_p (z-1)^p$$ (C.4)

$$(z-1)^{m_2+1} z^{m_4+1} = \sum_{p=0}^{\infty} c_p (z-x)^p$$ (C.5)

$$(z-1)^{m_2+1}(z-x)^{m_3+1} = \sum_{p=0}^{\infty} d_p z^p$$ (C.6)

**A useful identity**

We give here the following commutation identity [42]:

$$\langle \mathcal{O}_1 | \mathcal{O}_2(1)\mathcal{O}_3(x,\bar{x}) L_n | \mathcal{O}_4 \rangle - \langle \mathcal{O}_1 | L_n \mathcal{O}_2(1)\mathcal{O}_3(x,\bar{x}) | \mathcal{O}_4 \rangle$$
$$= \{(1-x^n)[x\partial_x + (n+1)h_3] + (h_4 - h_1) - n(h_2 + h_3)\} \langle \mathcal{O}_1 | \mathcal{O}_2(1)\mathcal{O}_3(x,\bar{x}) | \mathcal{O}_4 \rangle \quad \text{(C.7)}$$

where $\mathcal{O}_2$ and $\mathcal{O}_3$ are primary fields and $|\mathcal{O}_2\rangle$, $|\mathcal{O}_4\rangle$ are generic states. This commutator identity allows one to express insertions of Virasoro modes $L_n$ *inside* a correlation function in terms of differential operators acting *on* them.

# D    Rényi entropies for the critical Ising chain with mixed fixed BC

In this section, we will derive the bare and excited twist contributions to the second and third Rényi entropy in the critical Ising chain with fixed mixed BC $a = +, b = -$. In the Ising BCFT, the boundary field that interpolates between the corresponding conformal BC $|\pm\rangle$ is the operator $\psi_{2,1}^{(+-)}$, with conformal dimension $h_{2,1} = 1/2$. In the $\mathbb{Z}_N$ orbifold of this theory, the change in boundary conditions is implemented by the diagonal operator $\Psi_{2,1}^{(\alpha\beta)}$ defined as in (2.27).

The essential observation for the derivation of this section is that the space of conformal blocks is one-dimensional for the chiral correlators

$$\left\langle \Phi_{1,3} | \sigma_j^{[-k]}(1) \sigma_j^{[k]}(\eta) | \Phi_{1,3} \right\rangle \tag{D.1}$$

with $j \in \{\mathbf{1}, \phi_{1,3}\}$ in the $\mathbb{Z}_2$ and $\mathbb{Z}_3$ Ising orbifold CFTs. The result is obtained, as in the discussion of Section 3, from the fusion rules of these theories, found in [40] and [70]. These fusion rules also imply the leading singular behaviour of the conformal block around the points $\eta \in \{0, 1, \infty\}$. The corresponding exponents are given in Table 4.

|  | 0 | 1 | $\infty$ |
|---|---|---|---|
| $N = 2, j = \mathbf{1}$ | $-1$ | $-\frac{1}{16}$ | $-\frac{15}{16}$ |
| $N = 2, j = \phi_{1,3}$ | $-1$ | $-\frac{9}{16}$ | $-\frac{7}{16}$ |
| $N = 3, j = \mathbf{1}$ | $-1$ | $-\frac{1}{9}$ | $-\frac{8}{9}$ |
| $N = 3, j = \phi_{1,3}$ | $-1$ | $-\frac{4}{9}$ | $-\frac{5}{9}$ |

Table 4: Singular behaviour of the conformal block of (D.1) for different $N$ and twist field insertions $\sigma_j^{[k]}(\eta)$

In the $\eta \to 1$ channel, the exponent corresponds to the fusion

$$\sigma_j^{[k]} \times \sigma_j^{[-k]} \to \Phi_{\mathbf{1}} \tag{D.2}$$

for all the chiral correlators we are considering in this section. The diagonal operator $\Phi_{\mathbf{1}}$ is defined as in (2.10).

From the exponents around $\eta \to 0$ and $\eta \to 1$ we can determine the generic form of the conformal blocks for the four cases enumerated above to be:

$$f_j^{(N)}(\eta) = \eta^{-1}(1 - \eta)^{-2h_{\sigma_j}} P(\eta) \tag{D.3}$$

where $P(\eta)$ is a generic polynomial in $\eta$. Furthermore, taking into account the singular behaviour of $f_j^{(N)}(\eta)$ around $\eta \to \infty$, one can constrain its degree in all four cases to be $\leq 2$, so that we have:

$$f_j^{(N)}(\eta) = \eta^{-1}(1 - \eta)^{-2h_{\sigma_j}}(a_2\, \eta^2 + a_1\, \eta + a_0) \tag{D.4}$$

Around $\eta \to 1$, this function behaves as:

$$f_j^{(N)}(\eta) \sim (1 - \eta)^{-2h_{\sigma_j}} \left[ (a_2 + a_1 + a_0) + (a_2 - a_0)(1 - \eta) + a_2(1 - \eta)^2 + \ldots \right] \tag{D.5}$$

To find $a_i$, we will need to consider the first few results in the module of $\Phi_{\mathbf{1}}$ from the OPE of twist fields in the $\mathbb{Z}_N$ orbifold:

$$\sigma_j^{[k]}(\eta)\sigma_j^{[-k]}(1) = \Phi_{\mathbf{1}}(1) + \frac{2h_{\sigma_j}}{Nc}(1-\eta)^2\, T^{(0)}(1) + \dots \tag{D.6}$$

where $T^{(0)}(z) = L_{-2}^{(0)}\Phi_{\mathbf{1}}(z)$ is the SET of the chiral $\mathbb{Z}_N$ orbifold CFT. The corresponding structure constant has been determined by applying a $L_2^{(0)}$ from the left on both sides of the OPE, and power matching in $(1-\eta)$. Finally, the term at level 1 has vanished because the null vector $L_{-1}\mathbf{1} \equiv 0$ in the mother CFT induces the null-vectors $L_{-1}^{(r)}\Phi_{\mathbf{1}} \equiv 0$ in the orbifold.

Inserting (D.6) into (D.1) one finds, the coefficients

$$a_0 = a_2 = \frac{2h_{\sigma_j}}{Nc} \quad a_1 = 1 - 2a_0 \tag{D.7}$$

with which we fix the conformal blocks for all the cases presented in Table 4. We then use the block expansions for the mixed BC correlators to find:

$$\left\langle \sigma_j^{[k]}(z,\bar{z}) \right\rangle_N^{\alpha\beta} = g_+^{1-N} f_j^N(\eta) \tag{D.8}$$

where we have also used, notably, the results of (2.42) for the 1-point structure constant of twist fields. After mapping to the strip through (3.2), we find for $N=2$:

$$\langle \sigma_{\mathbf{1}}(\ell,\ell) \rangle_{\mathbb{S}_L}^{+-} = 2^{-5/2}\left(\frac{2L}{\pi}\right)^{-1/16} \frac{7 + \cos\frac{2\pi\ell}{L}}{\left(\sin\frac{\pi\ell}{L}\right)^{1/16}} \tag{D.9}$$

$$\langle \sigma_{\varepsilon}(\ell,\ell) \rangle_{\mathbb{S}_L}^{+-} = 2^{-5/2}\left(\frac{2L}{\pi}\right)^{-9/16} \frac{1 - 9\cos\frac{2\pi\ell}{L}}{\left(\sin\frac{\pi\ell}{L}\right)^{9/16}} \tag{D.10}$$

and $N=3$:

$$\langle \sigma_{\mathbf{1}}(\ell,\ell) \rangle_{\mathbb{S}_L}^{+-} = 3^{-2}\left(\frac{2L}{\pi}\right)^{-1/9} \frac{7 + 2\cos\frac{2\pi\ell}{L}}{\left(\sin\frac{\pi\ell}{L}\right)^{1/9}} \tag{D.11}$$

$$\langle \sigma_{\varepsilon}(\ell,\ell) \rangle_{\mathbb{S}_L}^{+-} = 3^{-2}\left(\frac{2L}{\pi}\right)^{-4/9} \frac{1 + 8\cos\frac{2\pi\ell}{L}}{\left(\sin\frac{\pi\ell}{L}\right)^{4/9}} \tag{D.12}$$

# E   Hypergeometric differential equation

The hypergeometric differential equation is canonically defined as:

$$\eta(\eta-1)f''(\eta) + [(a+b+1)\eta - c]f'(\eta) + ab\, f(\eta) = 0 \tag{E.1}$$

with the Riemann scheme:

| 0 | 1 | $\infty$ |
|---|---|---|
| 0 | 0 | $a$ |
| $1-c$ | $c-a-b$ | $b$ |

The solutions are constructed using the Gauss hypergeometric function $_2\mathrm{F}_1(a,b;c\mid\eta)$. Following the conventions of [94], we give a standard basis of fundamental solutions to (E.1) around the singular point $\eta = 0$:

$$
\begin{aligned}
I_1(\eta) &= {}_2\mathrm{F}_1(a,b;c\mid\eta) \\
I_2(\eta) &= \eta^{1-c}{}_2\mathrm{F}_1(b-c+1,a-c+1;2-c\mid\eta)
\end{aligned}
\tag{E.2}
$$

and around $\eta = 1$:

$$
\begin{aligned}
J_1(\eta) &= {}_2\mathrm{F}_1(a,b;a+b-c+1\mid 1-\eta) \\
J_2(\eta) &= (1-\eta)^{c-a-b}{}_2\mathrm{F}_1(c-b,c-a;c-a-b+1\mid 1-\eta)
\end{aligned}
\tag{E.3}
$$

The two bases of solutions are linearly related as

$$
I_i(\eta) = \sum_{j=1}^{2} P_{ij} J_j(\eta)
\tag{E.4}
$$

with the fusing matrix P

$$
\mathrm{P} = \begin{bmatrix}
\frac{\Gamma(c)\Gamma(d)}{\Gamma(c-a)\Gamma(c-b)} & \frac{\Gamma(c)\Gamma(-d)}{\Gamma(a)\Gamma(b)} \\
\frac{\Gamma(2-c)\Gamma(d)}{\Gamma(1-a)\Gamma(1-b)} & \frac{\Gamma(2-c)\Gamma(-d)}{\Gamma(1-c+a)\Gamma(1-c+b)}
\end{bmatrix}
\tag{E.5}
$$

and its inverse:

$$
\mathrm{P}^{-1} = \begin{bmatrix}
\frac{\Gamma(1-c)\Gamma(1-d)}{\Gamma(1-c+a)\Gamma(1-c+b)} & \frac{\Gamma(c-1)\Gamma(1-d)}{\Gamma(a)\Gamma(b)} \\
\frac{\Gamma(1-c)\Gamma(1+d)}{\Gamma(1-a)\Gamma(1-b)} & \frac{\Gamma(c-1)\Gamma(1+d)}{\Gamma(c-a)\Gamma(c-b)}
\end{bmatrix}
\tag{E.6}
$$

expressed in terms of Euler's Gamma function $\Gamma$, with $d = c - a - b$.

# F  Fusion rules in the $\mathbb{Z}_N$ orbifold

In [70] we have found compact expressions for the fusion numbers of the $\mathbb{Z}_N$ orbifold of a diagonal RCFT. They are given by:

$$
\begin{aligned}
\mathcal{N}^{[k_1\ldots k_N]}_{[i_1\ldots i_N],[j_1\ldots j_N]} &= \sum_{a,b=0}^{N-1} N^{k_1}_{i_{1+a},j_{1+b}}\ldots N^{k_N}_{i_{N+a},j_{N+b}}\,, \\
\mathcal{N}^{k^{(r)}}_{[i_1\ldots i_N],[j_1\ldots j_N]} &= \sum_{a=0}^{N-1} N^{k}_{i_{1+a},j_1}\ldots N^{k}_{i_{N+a},j_N}\,, \\
\mathcal{N}^{k^{(s)}}_{[i_1\ldots i_N],j^{(r)}} &= N^{k}_{i_1,j}\ldots N^{k}_{i_N,j}\,, \\
\mathcal{N}^{k^{(t)}}_{i^{(r)},j^{(s)}} &= \delta_{r+s,t}\, N^{k}_{ij}\,. \\
\mathcal{N}^{[k_1\ldots k_N]}_{i_{[p](r)}\,j_{[q](s)}} &= \delta_{p+q,0}\sum_{\ell=1}^{M} \frac{S_{i\ell}S_{j\ell}\cdot S_{k_1\ell}\ldots S_{k_N\ell}}{S_{1\ell}^{N}}\,, \\
\mathcal{N}^{k^{(t)}}_{i_{[p](r)}\,j_{[q](s)}} &= \frac{\delta_{p+q,0}}{N}\sum_{\ell=1}^{M}\left[\frac{S_{i\ell}S_{j\ell}S_{k\ell}^{N}}{S_{1\ell}^{N}} + \sum_{n=1}^{N-1}\omega^{np(r+s-t)}\frac{(P_{-n})_{i\ell}(P_n)_{j\ell}S_{k\ell}}{S_{1\ell}}\right]\,, \\
\mathcal{N}^{k^{[m](t)}}_{i_{[p](r)}\,j_{[q](s)}} &= \frac{\delta_{p+q,m}}{N}\sum_{\ell=1}^{M}\left[\frac{S_{i\ell}S_{j\ell}S_{k\ell}}{S_{1\ell}^{N}} + \sum_{n=1}^{N-1}\omega^{n(r+s-t)}\frac{(P^{\dagger}_{pn^{-1}})_{i\ell}(P^{\dagger}_{qn^{-1}})_{j\ell}(P_{mn^{-1}})_{k\ell}}{S_{1\ell}}\right]\,.
\end{aligned}
\tag{F.1}
$$

where $\omega = \exp\left(2\pi i/N\right)$, and $N_{ij}^k$, $S_{ij}$ are the fusion numbers and the modular $S$-matrix of the mother CFT. One also needs the matrix $P_n$ which is defined from

$$P_n = T^{-n/N} \cdot Q_n \cdot T^{[[-n^{-1}]]/N}, \qquad n \in \mathbb{Z}_N^\times, \tag{F.2}$$

where $T$ is the modular $T$ matrix of the mother CFT, $[[-n^{-1}]]$ denotes the inverse of $(-n)$ in $\mathbb{Z}_N^\times$, with $0 < [[-n^{-1}]] < N$, and $Q_n$ is the matrix representing the linear action of the modular map

$$\tau \mapsto q_n(\tau) = \frac{n\tau - (n[[-n^{-1}]] + 1)/N}{N\tau - [[-n^{-1}]]} \tag{F.3}$$

on the characters $\chi_j$ of the mother CFT.

# G   Derivation of differential equation in the $Z_3$ orbifold BCFT

We present in this section all the orbifold Ward identities and null-vector conditions necessary to derive the third order differential equation (3.45).

**The Ward identities**

**Ward 1** The correlator to integrate over is:

$$\langle\Phi_{12}|\, L_1^{(1)}\sigma_{\mathbf{1}}(1)T^{(1)}(z)\tilde{\sigma}_{\mathbf{1}}(\eta)L_{-1}^{(1)}\,|\Phi_{12}\rangle \tag{G.1}$$

with $(m_1, m_2, m_3, m_4) = (-1, 1/3, -1/3, -1)$, to find:

$$a_{0|1}\,\langle\Phi_{12}|\,(L_1^{(1)})^2\sigma_{\mathbf{1}}(1)\tilde{\sigma}_{\mathbf{1}}(\eta)L_{-1}^{(1)}\,|\Phi_{12}\rangle + a_{1|1}\,\langle\Phi_{12}|\,L_1^{(1)}L_0^{(1)}\sigma_{\mathbf{1}}(1)\tilde{\sigma}_{\mathbf{1}}(\eta)L_{-1}^{(1)}\,|\Phi_{12}\rangle = \atop d_{0|1}\,\langle\Phi_{12}|\,L_1^{(1)}\sigma_{\mathbf{1}}(1)\tilde{\sigma}_{\mathbf{1}}(\eta)L_{-1}^{(1)}L_{-1}^{(1)}\,|\Phi_{12}\rangle + d_{1|1}\,\langle\Phi_{12}|\,L_1^{(1)}\sigma_{\mathbf{1}}(1)\tilde{\sigma}_{\mathbf{1}}(\eta)L_0^{(1)}L_{-1}^{(1)}\,|\Phi_{12}\rangle \tag{G.2}$$

**Ward 2** The correlator to integrate over is:

$$\langle\Phi_{12}|\,\sigma_{\mathbf{1}}(1)T^{(1)}(z)\tilde{\sigma}_{\mathbf{1}}(\eta)L_{-1}^{(1)}L_{-1}^{(1)}\,|\Phi_{12}\rangle \tag{G.3}$$

with $(m_1, m_2, m_3, m_4) = (-1, 1/3, -1/3, -1)$ to find:

$$a_{0|2}\,\langle\Phi_{12}|\,L_1^{(1)}\sigma_{\mathbf{1}}(1)\tilde{\sigma}_{\mathbf{1}}(\eta)\left(L_{-1}^{(1)}\right)^2\,|\Phi_{12}\rangle = d_{0|2}\,\langle\Phi_{12}|\,\sigma_{\mathbf{1}}(1)\tilde{\sigma}_{\mathbf{1}}(\eta)\left(L_{-1}^{(1)}\right)^3\,|\Phi_{12}\rangle$$

$$+d_{1|2}\,\langle\Phi_{12}|\,\sigma_{\mathbf{1}}(1)\tilde{\sigma}_{\mathbf{1}}(\eta)L_0^{(1)}\left(L_{-1}^{(1)}\right)^2\,|\Phi_{12}\rangle + d_{2|2}\,\langle\Phi_{12}|\,\sigma_{\mathbf{1}}(1)\tilde{\sigma}_{\mathbf{1}}(\eta)L_1^{(1)}\left(L_{-1}^{(1)}\right)^2\,|\Phi_{12}\rangle \tag{G.4}$$

$$+ d_{3|2}\,\langle\Phi_{12}|\,\sigma_{\mathbf{1}}(1)\tilde{\sigma}_{\mathbf{1}}(\eta)L_2^{(1)}\left(L_{-1}^{(1)}\right)^2\,|\Phi_{12}\rangle$$

**Ward 3** The correlator to integrate over is

$$\langle\Phi_{12}|\,L_1^{(1)}L_1^{(1)}\sigma_{\mathbf{1}}(1)T^{(1)}(z)\tilde{\sigma}_{\mathbf{1}}(\eta)\,|\Phi_{12}\rangle \tag{G.5}$$

with $(m_1, m_2, m_3, m_4) = (-1, 1/3, -1/3, -1)$ to find:

$$d_{0|3}\,\langle\Phi_{12}|\,\left(L_1^{(1)}\right)^2\sigma_{\mathbf{1}}(1)\tilde{\sigma}_{\mathbf{1}}(\eta)L_{-1}^{(1)}\,|\Phi_{12}\rangle = a_{0|3}\,\langle\Phi_{12}|\,\left(L_1^{(1)}\right)^3\sigma_{\mathbf{1}}(1)\tilde{\sigma}_{\mathbf{1}}(\eta)\,|\Phi_{12}\rangle$$

$$+a_{1|3}\,\langle\Phi_{12}|\,\left(L_1^{(1)}\right)^2L_0^{(1)}\sigma_{\mathbf{1}}(1)\tilde{\sigma}_{\mathbf{1}}(\eta)\,|\Phi_{12}\rangle + a_{2|3}\,\langle\Phi_{12}|\,\left(L_1^{(1)}\right)^2L_{-1}^{(1)}\sigma_{\mathbf{1}}(1)\tilde{\sigma}_{\mathbf{1}}(\eta)\,|\Phi_{12}\rangle \tag{G.6}$$

$$+ a_{3|3}\,\langle\Phi_{12}|\,\left(L_1^{(1)}\right)^2L_{-2}^{(1)}\sigma_{\mathbf{1}}(1)\tilde{\sigma}_{\mathbf{1}}(\eta)\,|\Phi_{12}\rangle$$

**Ward 4** The correlator to integrate over is:

$$\langle \Phi_{12}| \, \sigma_{\mathbf{1}}(1) T^{(2)}(z) \tilde{\sigma}_{\mathbf{1}}(\eta) L_{-1}^{(1)} |\Phi_{12}\rangle \tag{G.7}$$

with $(m_1, m_2, m_3, m_4) = (0, -1/3, 1/3, -2)$ so we find:

$$d_{0|4} \, \langle \Phi_{12}| \, \sigma_{\mathbf{1}}(1) \tilde{\sigma}_{\mathbf{1}}(\eta) L_{-2}^{(2)} L_{-1}^{(1)} |\Phi_{12}\rangle + d_{1|4} \, \langle \Phi_{12}| \, \sigma_{\mathbf{1}}(1) \tilde{\sigma}_{\mathbf{1}}(\eta) L_{-1}^{(2)} L_{-1}^{(1)} |\Phi_{12}\rangle +$$
$$d_{2|4} \, \langle \Phi_{12}| \, \sigma_{\mathbf{1}}(1) \tilde{\sigma}_{\mathbf{1}}(\eta) L_{0}^{(2)} L_{-1}^{(1)} |\Phi_{12}\rangle + d_{3|4} \, \langle \Phi_{12}| \, \sigma_{\mathbf{1}}(1) \tilde{\sigma}_{\mathbf{1}}(\eta) L_{1}^{(2)} L_{-1}^{(1)} |\Phi_{12}\rangle = 0 \tag{G.8}$$

**Ward 5** The correlator to integrate over is:

$$\langle \Phi_{12}| \, L_{1}^{(1)} T^{(2)}(z) \sigma_{\mathbf{1}}(1) \tilde{\sigma}_{\mathbf{1}}(\eta) |\Phi_{12}\rangle \tag{G.9}$$

with $(m_1, m_2, m_3, m_4) = (-2, -1/3, 1/3, 0)$ so we find:

$$a_{0|5} \, \langle \Phi_{12}| \, L_{1}^{(1)} L_{2}^{(2)} \sigma_{\mathbf{1}}(1) \tilde{\sigma}_{\mathbf{1}}(\eta) |\Phi_{12}\rangle + a_{1|5} \, \langle \Phi_{12}| \, L_{1}^{(1)} L_{1}^{(2)} \sigma_{\mathbf{1}}(1) \tilde{\sigma}_{\mathbf{1}}(\eta) |\Phi_{12}\rangle +$$
$$a_{2|5} \, \langle \Phi_{12}| \, L_{1}^{(1)} L_{0}^{(2)} \sigma_{\mathbf{1}}(1) \tilde{\sigma}_{\mathbf{1}}(\eta) |\Phi_{12}\rangle + a_{3|5} \, \langle \Phi_{12}| \, L_{1}^{(1)} L_{-1}^{(2)} \sigma_{\mathbf{1}}(1) \tilde{\sigma}_{\mathbf{1}}(\eta) |\Phi_{12}\rangle = 0 \tag{G.10}$$

**Ward 6** The correlator to integrate over is:

$$\langle \Phi_{12}| \, \sigma_{\mathbf{1}}(1) T^{(2)}(z) \tilde{\sigma}_{\mathbf{1}}(\eta) L_{-1}^{(1)} |\Phi_{12}\rangle \tag{G.11}$$

with $(m_1, m_2, m_3, m_4) = (-1, -1/3, 1/3, -1)$ to find:

$$a_{0|6} \, \langle \Phi_{12}| \, L_{1}^{(2)} \sigma_{\mathbf{1}}(1) \tilde{\sigma}_{\mathbf{1}}(\eta) L_{-1}^{(1)} |\Phi_{12}\rangle = d_{0|6} \, \langle \Phi_{12}| \, \sigma_{\mathbf{1}}(1) \tilde{\sigma}_{\mathbf{1}}(\eta) L_{-1}^{(2)} L_{-1}^{(1)} |\Phi_{12}\rangle$$
$$+ d_{1|6} \, \langle \Phi_{12}| \, \sigma_{\mathbf{1}}(1) \tilde{\sigma}_{\mathbf{1}}(\eta) L_{0}^{(2)} L_{-1}^{(1)} |\Phi_{12}\rangle + d_{2|6} \, \langle \Phi_{12}| \, \sigma_{\mathbf{1}}(1) \tilde{\sigma}_{\mathbf{1}}(\eta) L_{1}^{(2)} L_{-1}^{(1)} |\Phi_{12}\rangle \tag{G.12}$$

**Ward 7** The correlator to integrate over is:

$$\langle \Phi_{12}| \, \sigma_{\mathbf{1}}(1) T^{(1)}(z) \tilde{\sigma}_{\mathbf{1}}(\eta) L_{-1}^{(2)} |\Phi_{12}\rangle \tag{G.13}$$

with $(m_1, m_2, m_3, m_4) = (-1, 1/3, -1/3, -1)$ to find:

$$a_{0|7} \, \langle \Phi_{12}| \, L_{1}^{(1)} \sigma_{\mathbf{1}}(1) \tilde{\sigma}_{\mathbf{1}}(\eta) L_{-1}^{(2)} |\Phi_{12}\rangle = d_{0|7} \, \langle \Phi_{12}| \, \sigma_{\mathbf{1}}(1) \tilde{\sigma}_{\mathbf{1}}(\eta) L_{-1}^{(1)} L_{-1}^{(2)} |\Phi_{12}\rangle$$
$$+ d_{1|7} \, \langle \Phi_{12}| \, \sigma_{\mathbf{1}}(1) \tilde{\sigma}_{\mathbf{1}}(\eta) L_{0}^{(1)} L_{-1}^{(2)} |\Phi_{12}\rangle + d_{2|7} \, \langle \Phi_{12}| \, \sigma_{\mathbf{1}}(1) \tilde{\sigma}_{\mathbf{1}}(\eta) L_{1}^{(1)} L_{-1}^{(2)} |\Phi_{12}\rangle \tag{G.14}$$

**The null-vector conditions**

$$L_{-1}^{(1)} L_{-1}^{(2)} |\Phi_{12}\rangle = \frac{1}{2} \left[ 3g L_{-2}^{(0)} - \left( L_{-1}^{(0)} \right)^2 \right] |\Phi_{12}\rangle \tag{G.15}$$

$$\langle \Phi_{12}| \, L_{1}^{(1)} L_{1}^{(2)} = \langle \Phi_{12}| \, \frac{1}{2} \left[ 3g L_{2}^{(0)} - \left( L_{1}^{(0)} \right)^2 \right] \tag{G.16}$$

$$2 \, L_{-1}^{(0)} L_{-1}^{(2)} L_{-1}^{(1)} |\Phi_{12}\rangle = \left[ 3g L_{-1}^{(0)} L_{-2}^{(0)} - \left( L_{-1}^{(0)} \right)^3 \right] |\Phi_{12}\rangle \tag{G.17}$$

$$2 \, L_{-1}^{(0)} L_{-1}^{(2)} L_{-1}^{(1)} |\Phi_{12}\rangle = \left[ 3g L_{-1}^{(1)} L_{-2}^{(2)} - \left( L_{-1}^{(1)} \right)^3 \right] |\Phi_{12}\rangle \tag{G.18}$$

$$2 \, \langle \Phi_{12}| \, L_{1}^{(0)} L_{1}^{(2)} L_{1}^{(1)} = \langle \Phi_{12}| \left[ 3g L_{2}^{(0)} L_{1}^{(0)} - \left( L_{1}^{(0)} \right)^3 \right] \tag{G.19}$$

$$2 \, \langle \Phi_{12}| \, L_{1}^{(0)} L_{1}^{(2)} L_{1}^{(1)} = \langle \Phi_{12}| \left[ 3g L_{2}^{(2)} L_{1}^{(1)} - \left( L_{1}^{(1)} \right)^3 \right] \tag{G.20}$$

By removing from this linear system of 13 equations all terms containing modes $L_n^{(r)}$ with $r \neq 0$, one indeed obtains (3.45).

# H Numerical implementation of the Frobenius method

We want to find a basis of solutions to the differential equation (3.30) that converge on the entire range of interest - the unit circle $|\eta| = 1$.

The Fuchsian ODE (3.30) has singular points $0, 1, \infty$. The solutions around $\eta = 0$ and $\eta = 1$ converge on the disks $|\eta| < 1$ and $|\eta - 1| < 1$ respectively. Thus, only a portion of the unit semicircle, namely $0 < \text{Arg}(\eta) < \pi/3$, is contained in the convergence disk around $\eta = 1$. We can circumvent this problem by observing that the solutions around $\eta = \infty$ can be convergent on the whole unit circle $|\eta| = 1$. Even better, we can implement the change of variable

$$\eta \mapsto \frac{1 + u}{2u}, \qquad \partial_\eta \mapsto -2u^2 \partial_u. \tag{H.1}$$

so that the new ODE, in the variable $u$ has singular points at $u = 0, 1, -1$. The original unit circle $|\eta| = 1$ is mapped to $|u - 1/3| = 2/3$, which is contained in the convergence disk $|u| < 1$. Hence, applying the Froebenius method, and expressing the solutions around $u = 1$ in terms of those around $u = 0$ will give the appropriate numerical evaluation of the desired values of $\eta$.

Now, as explained in [42], a convenient way of finding power series solutions around a point $u = u_0$ is to rewrite the differential equation (3.30) in terms of the operator $\theta = (u - u_0)\partial_u$, which satisfies:

$$(u - u_0)^n \partial_u^n = \prod_{k=0}^{n-1} (\theta - k) \tag{H.2}$$

Most importantly, we have that any polynomial $P(\theta)$ satisfies:

$$P(\theta)(u - u_0)^r = P(r)(u - u_0)^r \tag{H.3}$$

For $u_0 = 0$, we can then rewrite the equation as:

$$\left[ \sum_{i=0}^{8} u^i P_i(\theta) \right] = 0 \tag{H.4}$$

where:

$$
\begin{aligned}
P_0(\theta) &= 250\theta^4 - 125\theta^3 - 130\theta^2 - \theta + 6 \\
P_1(\theta) &= -\theta \left( -2125\theta^2 + 450\theta + 997 \right) - 174 \\
P_2(\theta) &= -250\theta^4 + 125\theta^3 + 130\theta^2 - \left( 750\theta^3 + 1250\theta^2 - 2415\theta + 4264 \right)\theta + \theta - 3123 \\
P_3(\theta) &= -\theta \left( 4250\theta^2 + 5925\theta + 6016 \right) + \theta \left( -2125\theta^2 + 450\theta + 997 \right) - 8511 \\
P_4(\theta) &= 5\,\theta \left( 150\theta^3 + 575\theta^2 + 93\theta - 332 \right) + \theta \left( 750\theta^3 + 1250\theta^2 - 2415\theta + 4264 \right) - 6000 \quad \text{(H.5)} \\
P_5(\theta) &= 2125\,\theta \left( \theta^2 + 3\theta + 2 \right) + \theta \left( 4250\theta^2 + 5925\theta + 6016 \right) \\
P_6(\theta) &= -250\,\theta \left( \theta^3 + 6\theta^2 + 11\theta + 6 \right) - 5\theta \left( 150\theta^3 + 575\theta^2 + 93\theta - 332 \right) \\
P_7(\theta) &= -2125\,\theta \left( \theta^2 + 3\theta + 2 \right) \\
P_8(\theta) &= 250\,\theta \left( \theta^3 + 6\theta^2 + 11\theta + 6 \right)
\end{aligned}
$$

We now seek power series solutions around $u = 0$ of the form:

$$I_i(u) = u^{r_i} \sum_{n=0}^{\infty} a_n u^n \quad \text{with} \quad a_0 = 1 \tag{H.6}$$

where the $r_i$ are the roots of the *characteristic polynomial* $P_0(r)$:

$$r_1 = -3/10 \quad r_2 = 1 \quad r_3 = 1/5 \quad r_4 = -2/5 \tag{H.7}$$

and are the same as the exponents around $\infty$ in Table 1.

By substituting the ansatz (H.6) in the differential equation and employing the identity (H.3) we find the following recursion relations for the coefficients $a_n$ of the solution $I_i(u)$:

$$P_0(r_i + n)a_n = -\sum_{i=1}^{min\{n,8\}} a_{n-i} P_i(r_i + n - i) \qquad a_0 = 1 \tag{H.8}$$

The four series found in this way converge for $|u| < 1$ and can be evaluated numerically to arbitrary precision.

We note, at this point, that the solution corresponding to $r_4$ is unphysical, since it corresponds, according to Table 1, to the presence in the operator algebra of the theory of a composite twist field formed with a primary operator that is outside the Kac table, i.e. not present in the $\mathcal{M}(6,5)$ CFT. This suggests that the physical space of conformal blocks is actually three-dimensional, and thus, that there should be a third order differential equation satisfied by the excited twist correlator in this setup.

One should now repeat the above computation for the solutions $J_j(u)$ around $u = 1$, since these are the ones that appear in the block expansion (3.4) of BCFT correlators. The recursion relation takes the same form as in (H.8), with different roots:

$$\lambda_1 = -1/2 \quad \lambda_2 = 9/10 \quad \lambda_3 = 23/10 \quad \lambda_4 = 23/10 + 2 \tag{H.9}$$

A slight complication appears in this case because two of the roots of the corresponding characteristic polynomial differ by an integer, that is, $r_4 = r_3 + 2$. This will lead to the truncation of the corresponding recursion relations (H.8) for $r_3$ because at $n = 2$, the coefficient $P_0(r_3 + 2) = 0$. A good basis of solutions in this case is $\{J_1(u), J_2(u), J_3^{(k)}(u), J_4(u)\}$, where:

$$J_i(u) = (1-u)^{r_i} \sum_{i=0}^{\infty} a_n(1-u)^n$$

$$J_3^{(k)}(u) = (1-u)^{r_3}[a_0 + a_1(1-u)] + kJ_4(u) \tag{H.10}$$

where $k$ is a free parameter and the value we choose for it should not change the final result for the physical correlator. We have chosen to set it to $k_0 = 0.04428171795178596$ and define $J_3(u) = J^{(k_0)}(u)$.

The reason for this choice becomes apparent when one looks at our solution for the fusing matrix $\mathbf{M}$:

$$\mathbf{M} = \begin{pmatrix} 0.207411 & 0.393808 & 0.152178 & 0 \\ 1.356 & -2.30281 & -0.444933 & 0 \\ 7.70383 & 71.6374 & -22.7841 & 0 \\ -8986.23 & -19156.8 & 7211.61 & 5800.8 \end{pmatrix} \tag{H.11}$$

which relates the bases of conformal blocks around $u = 1$ and $u = 0$ as:

$$J_i(u) = \sum_j M_{ij} I_j(u) \tag{H.12}$$

To obtain this solution, we have generated a linear system of equations for the unknown $M_{ij}$ from the evaluation of the above relations at different points $\{u_i\}$ in the interval $0 < u < 1$ (where both sets of solutions converge). In this context, the parameter $k_0$ was tuned so that the block $J_3(u)$ does not depend on the unphysical solution $I_3(u)$ around $u = 0$. Furthermore, since the matrix elements $(M^{-1})_{i4} = 0$ can be readily checked to vanish for $i \in \{1, 2, 3\}$, we can conclude that $\{J_1(u), J_2(u), J_3(u)\}$ form the physical three-dimensional basis of conformal blocks around $u = 1$.

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
