# Peer review of "Rényi entropies for one-dimensional quantum systems with mixed boundary conditions"

_SciPost Physics_

## Round 2 · Referee Report · Anonymous (Referee 1) · 2025-5-25

Strengths
- In a Mathematical Physics context this paper provides a high level of rigour and precision, exploiting conformal field theory techniques to obtain exact results.
- It provides a quite general method to compute entanglement measures in CFT for generic boundary conditions, paving the way for further analytic or at least partially analytic results. -It is well written and very detailed.
Weaknesses
Report
The authors consider the entanglement corresponding to a segment starting at one of the boundaries. It is known form the literature that this is related to the one-point function of a symmetry field, also called a twist field. The main quantity of interest is a scaling function that interpolates between the values of the g-function corresponding to each of the boundary conditions. The authors employ advanced features of CFT such as the presence of null vectors, leading to differential equations for certain correlators which can be solved exactly in some cases, or the operator product expansion of fields, which is exploited for instance to consider contributions from higher order terms in this expansion. Indeed, the computations and presented numerical work demonstrate that these corrections can play a very significant role at finite volume.
As I wrote before, the computations are presented in detail and the techniques explained clearly so that other colleagues in the field will be able to take this work at starting point for learning such techniques and possibly expanding their use to other entanglement measures/CFTs.
My view is that the paper should be published in SciPost with minor changes.
Requested changes
I have just a couple of comments, the main one is about literature.
- I think there is a small issue in the abstract where the last paragraph seems to start abruptly. I think the 2nd and 3rd paragraph were meant to be a single paragraph?
- The authors use twist fields as their leading method for performing calculations. The first time they appear is in their equation (2.1) where they identify the Rényi entropies as proportional to the logarithm of the one-point function of a twist field with dimension (2.3). Before (2.1) they cite [74] for the twist fields and before (2.3) they cite [2,44,45]. However none of these references includes the use or definition of the twist field that they authors use here, neither do they contain the dimension (2.3).
Let me focus on reference [74] and I will refer here to the version available on ArXiv https://arxiv.org/pdf/hep-th/0405152.
In this reference entanglement measures are defined in terms of a field associated to conical singularities. The conformal dimension of this field is computed and one can see it in many places: towards the end of page 7, just before equation (12), again before equation (16), and again after equation (20). In all of these cases the dimension is not the formula (2.3) in the current paper. It is actually formula (2.3) divided by N. The fact that this is the case is the reason why also in reference [74] we are told just before equation (16) that: "we conclude that the renormalised (partition function) behaves (apart from a possible overall constant) under scale and conformal transformations identically to the nth power of two-point function of a primary operator \Phi_n". Thus, the authors of [74] consider the Nth power of their correlators in order to get the correct scaling of the Rényi entropies.
This is not what the authors of the current paper are doing. They are not taking the Nth power of the one-point function in (2.1) and they are considering a field of dimension (2.3). This field is not longer a field associated to conical singularities in the sense of [74] but a field associated to a symmetry (hence, a twist field). The distinction is subtle, but not insignificant. The symmetry is cyclic permutation of copies and this viewpoint and definition (even the terminology "twist field") were first used in the context of entanglement in the paper https://arxiv.org/abs/0706.3384. This viewpoint is also the one largely adopted by the community ever since.
Although https://arxiv.org/abs/0706.3384 has a focus on massive theories, it does present the derivation of the conformal dimension in CFT giving the formula (2.3). It gives that formula because, crucially, in a replica theory the central charge is Nc rather than c and this N is what gives the missing N in the conformal dimension so that the partition function becomes a correlator, rather than a power of a correlator. This is very much the same the authors of the current paper do when they write their equation (2.7) for the dimension of their field Phi.
For all these reasons, the paper https://arxiv.org/abs/0706.3384 must be cited.
- One suggestion that comes to mind is that there is another known function that interpolates between the values of the g-functions for two distinct boundary conditions. This is computed in the context of the (boundary) thermodynamic Bethe ansatz (see e.g. https://arxiv.org/pdf/1809.05705) and it could be interesting to investigate/compare these two functions.
Recommendation
Publish (easily meets expectations and criteria for this Journal; among top 50%)
We would like to thank the referee for carefully reading our manuscript.
1. We have corrected the issue with the abstract!
-
We fully agree with the comment and we will update the references accordingly to properly reflect the distinction between the fields associated with conical singularities (as in [74]) and the twist fields associated with cyclic permutation symmetry. In particular, we will explicitly cite the paper https://arxiv.org/abs/0706.3384, which clearly presents the derivation of the conformal dimension in the replica CFT framework and has indeed been foundational for the community’s understanding of twist fields in entanglement studies.
-
We thank the referee for this suggestion, it may be interesting to investigate this relation in some future work!

Author: Andrei Rotaru on 2025-09-11 [id 5805]
(in reply to Report 2 on 2025-06-05)We would like to thank the referee for carefully reading our manuscript. Here’s our reply to the comments and suggestions in the order in which they appear in the report.
In our case, the UV cutoff is explicitly given by the lattice spacing a, which is not introduced as an abstract regulator but rather is a physical parameter intrinsic to the underlying lattice model. Accordingly, we have been very careful and consistent in our treatment of this cutoff throughout the manuscript. In particular, the role of the lattice spacing and its implications for the correlators of twist operators are discussed in detail around equations (2.10) and (2.12). We clarify there how the cutoff appears in various expressions, and why in some cases it can be absorbed or dropped due to normalization or scaling arguments, without loss of generality or inconsistency.
We have clarified this point in the revised version of the manuscript (see the conclusion) to more clearly articulate the rationale and benefits of the orbifold framework.
The condition $h_{13}>0$ is always satisfied when working within a unitary conformal field theory, which is the physically relevant case for entanglement entropy computations. When $h_{13}\leq 0$, we are effectively dealing with a non-unitary CFT, and in such theories the very definition and interpretation of entanglement entropy becomes problematic. For this reason, we do not consider this regime physically meaningful in the context of our work.
That said, from a purely formal standpoint, the universal scaling function F we define is based on substracting the leading divergence of the entropy that arises as the twist field approaches the boundary. This leading behavior is captured by the $\log\sin$ term in Eq. (1.4), under the assumption that the identity is the most relevant operator in the boundary OPE.
When $h_{13}\leq 0$, this assumption breaks down: the identity operator is no longer the dominant contribution in the OPE, and subleading operators become more relevant. As a result, the subtraction no longer removes the leading divergence, and the function F appears to diverge as the boundary is approached.
To treat the case $h_{13}\leq 0$, one should redefine F by subtracting the correct leading divergence in that regime (which would simply amount to putting the appropriate prefactor in front of the log sin term). However, since our focus is on unitary theories relevant to entanglement entropy, we restrict our analysis to the $h_{13}>0$ regime, where the current definition of F is both meaningful and well-behaved.
We have added some clarifying remarks in the beginning of Sec 2.1, and after eq. (2.25), and a comment on the behaviour of the solutions of the ODE in Sec 4.1.
While the method proposed by the referee may be valid and useful within the continuum field theory setup (with a chosen regularization), it is not applicable to our setting without making strong assumptions about how the lattice model flows to the continuum. In particular, the regularization introduced in arXiv:1406.4167, while elegant in the continuum, does not capture the rich structure of lattice-specific corrections we observe.
Anonymous on 2025-09-13 [id 5809]
(in reply to Andrei Rotaru on 2025-09-11 [id 5805])Thank you for the clarification.
I am broadly satisfied with the answers.
However, there remain aspects of the discussion concerning the twist operator that I am not yet convinced by.
Below is my view:
Regularization is indispensable when defining a CFT on the replica manifold.
The authors argue that this regularization is different from the lattice spacing,
but the two should rather be identified.
When describing the continuum limit of a lattice model in a field theory,
the UV cutoff parameter that arises from regularization on the CFT side corresponds to the lattice spacing on the lattice side.
The claim that one “defines the twist operator in the lattice model” amounts only to establishing a correspondence between the parameters on the two sides.
The twist operator itself is defined in field theory as the endpoint of the Z_n symmetry defect,
and by definition, its construction involves a cutoff parameter.
What the authors are actually doing is to read off the correspondence between this cutoff parameter and the lattice spacing.
If one regards the regularization and the lattice spacing as distinct,
then the authors’ formulas should explicitly contain an additional,
physically meaningful parameter—namely,
the regularization parameter of the replica manifold.
This parameter is not something that can be eliminated at will.
Additionally, in field theory, the subleading in the regularization parameter is actually model-dependent—unlike the leading term—and therefore provides a meaningful basis for comparison with the lattice model.
To summarize, in field theory, a twist operator is not something that can be inserted by hand at will, but a mathematically constructed object defined to be consistent with the theory, and calculations should be carried out in accordance with that definition.

---

## Round 2 · Referee Report · Anonymous (Referee 2) · 2025-6-5

Strengths
-
This work establishes a new approach to Renyi entropy in finite systems by deriving a differential equation for correlation functions involving null vectors and twist operators on the disk.
-
This work provided a strong candidate for the correction to the Renyi entropy, which is necessary to resolve the discrepancies with numerical results observed in previous works.
Weaknesses
-
The method used in this work is essentially a slight modification of arXiv:1709.09270, and thus lacks technical novelty.
-
The analysis is restricted to small values of $N$, so the entanglement entropy (i.e. the von-Neumann limit of the Renyi entropy) cannot be computed.
Report
Requested changes
Comments:
-
The treatment of the UV cutoff parameter $a$ is generally sloppy. In some places, the correlators of the twist operators include the UV cutoff, while in others they do not, leading to inconsistency. In general, the definition of twist operators involves regularizing the conical singularity, therefore, correlators of twist operators should always include the cutoff parameter (see, for example, hep-th/0006196). In any case, the treatment of regularization should be made more precise.
-
The paper claims an advantage of using twist operators in the computation of Renyi entropy, but (at least within the setup considered in this work) no such advantage is apparent. The four-point function of null vectors and twist operators obeys a certain differential equation, which simplifies to a tractable level for $N = 2, 3$. This is a key point of the paper. Let us consider this problem in the unfolded picture. The resulting $2N$-point function obeys the standard BPZ equations. (Since the $2N$ points are symmetrically arranged, it should not be too difficult to reduce the dependence on the moduli parameters to a single variable.) While solving the BPZ equations for general $N$ is challenging, it is certainly feasible for $N = 2, 3$. Thus, the difficulty level of the problem appears to be comparable in both the folded and unfolded pictures. If my understanding here is lacking, and there is indeed a fundamental difference in the computational difficulty between the two approaches, then I would appreciate a clearer explanation of that distinction. Just to be clear, I am not claiming that the folded picture computation is not meaningless. I do recognize the value in developing alternative computational methods. However, if the authors wish to assert a clear advantage of their approach (especially in the systems treated in this paper), they should state the grounds for such a claim more clearly.
-
Under equation (2.25), the condition $h_{13} > 0$ is imposed. What happens when $h_{13} \leq 0$? At first glance, it seems counterintuitive that the solution to a hypergeometric differential equation would change discontinuously at such a point, so further explanation would be helpful.
Suggestion:
- The coefficients of the composite twist operator corrections are determined via fitting. Is it not possible to compute these coefficients analytically? Of course, this would require specifying the regularization. A natural and simple regularization choice is introduced in arXiv:1406.4167. Using this regularization, one should be able to fix the coefficients analytically. If the results then agree with numerical data, that would be quite interesting.
Recommendation
Publish (meets expectations and criteria for this Journal)

---

## Editorial Decision

unknown